# An Efficient Recipe for Long Context Extension via Middle-Focused Positional Encoding

**Tong Wu**
wutong1@bigai.ai

**Yanpeng Zhao**
zhaoyanpeng@bigai.ai

**Zilong Zheng**[✉]
zlzheng@bigai.ai

State Key Laboratory of General Artificial Intelligence, BIGAI, Beijing, China
[✉] Corresponding author.

## Abstract

Recently, many methods have been developed to extend the context length of pre-trained large language models (LLMs), but they often require fine-tuning at the target length ($\gg 4K$) and struggle to effectively utilize information from the middle part of the context. To address these issues, we propose **C**ontinuity-**R**elativity ind**E**xing with g**A**ussian **M**iddle (`CREAM`), which interpolates positional encodings by manipulating position indices. Apart from being simple, `CREAM` is training-efficient: it only requires fine-tuning at the pre-trained context window (*e.g.*, Llama 2-4K) and can extend LLMs to a much longer target context length (*e.g.*, 256K). To ensure that the model focuses more on the information in the middle, we introduce a truncated Gaussian to encourage sampling from the middle part of the context during fine-tuning, thus alleviating the "Lost-in-the-Middle" problem faced by long-context LLMs. Experimental results show that `CREAM` successfully extends LLMs to the target length for both Base and Chat versions of `Llama2-7B` with "Never Miss A Beat". Our code is publicly available at https://github.com/bigai-nlco/cream.

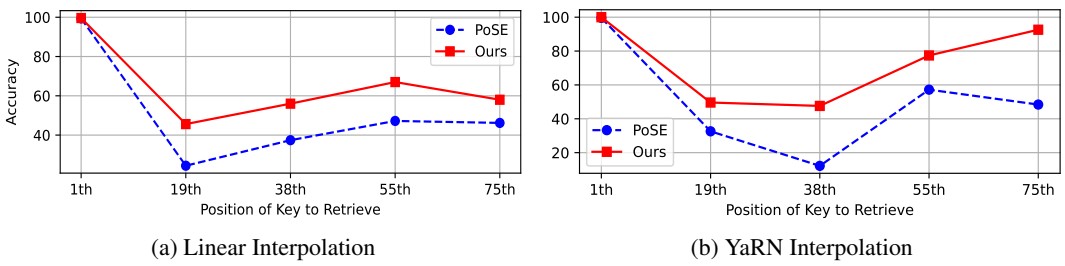

(a) Linear Interpolation  (b) YaRN Interpolation

Figure 1: Results of applying different position interpolation methods to the "Lost-in-the-Middle" task on `CREAM` and PoSE [Zhu et al., 2023]. We can see that `CREAM` outperforms PoSE [Zhu et al., 2023] at every position, with a particularly improvement in the middle.

## 1 Introduction

Transformer-based Large Language Models (LLMs) are typically pre-trained with a fixed context window size, *e.g.*, 4K tokens in Touvron et al. [2023a]. However, many downstream applications, including in-context learning [Huang et al., 2023, Li et al., 2023a] and LLM agents [Qian et al., 2023, Zheng et al., 2023] necessitate the processing of significantly longer contexts, *e.g.*, up to 256K tokens. Recent works have proposed promising approaches that efficiently extend the context window of

38th Conference on Neural Information Processing Systems (NeurIPS 2024).

pre-trained LLMs by interpolating Positional Encodings (PEs) [Chen et al., 2023, Peng and Quesnelle, 2023, Peng et al., 2023, Xiong et al., 2023, Zhang et al., 2024] with a short period of fine-tuning. Unlike other techniques such as efficient transformer [Tworkowski et al., 2024, Munkhdalai et al., 2024] and memory augmentation [Tan et al., 2024], PE-based methods do not necessitate alterations to the model's architecture or the incorporation of supplementary modules. Consequently, PE-based methods offer the advantages of straightforward implementation and rapid adaptation, making them a practical solution for extending the operational range of LLMs in tasks involving larger context windows.

Despite the simplicity and effectiveness, existing PE-based methods exhibit two significant limitations. **First,** prior approaches, such as positional interpolation [Chen et al., 2023], still require fine-tuning on the target context window size, which imposes a substantial *computational overhead* [Zhu et al., 2023]. **Secondly,** though some PE methods have demonstrated potential in handling extremely long sequences, as evidenced by low sliding window perplexity scores, their performance deteriorates notably in "in-the-middle" scenarios [Liu et al., 2024]. Specifically, when the model is required to accurately retrieve and process content located in the middle of an extended context, there is a marked drop in performance on the extended window size (Figure 1 and Figure 3).

These observations and insights underscore a fundamental question: *Can we extend the context window size of pre-trained LLMs **efficiently** while simultaneously optimizing their **effectiveness** in processing "in-the-middle" content?*

To answer the above question, we propose CREAM, namely **C**ontinuity-**R**elativity ind**E**xing with g**A**ussian **M**iddle. CREAM is a novel PE-based fine-tuning recipe that shows both efficiency in fine-tuning and effectiveness in enhanced middle content understanding. Our key insights lie in manipulating the positional indices of long target sequences to produce shorter ones within the pre-trained context window size (Figure 2).

In Section 2.1, we summarize two crucial ingredients of effective positional indices: continuity that produces densely connected positional indices and relativity that reveals the long-range dependencies between fragments. CREAM is a recipe designed with the best of both worlds by introducing two indexing strategies for continuity and relativity, respectively (Section 2.2). Besides, to alleviate the "Lost-in-the-Middle" challenge, we introduce truncated Gaussian distribution for middle segment sampling, enabling the LLM to prioritize the information in the middle positions, even when performing positional interpolation within the pre-trained context window size.

In Section 3, we conduct comprehensive experiments to demonstrate the efficiency and effectiveness of CREAM. We continually pre-trained on `Llama 2-7B` with CREAM for a short period and extend the context window size from 4K to up to 256K. Furthermore, we instruction tuning on `Llama 2-7B-Chat` with CREAM for 100 steps and obtain promising results. We highlight our empirical advantages as:

1. CREAM can not only fine-tune within the pre-training context window size, but also alleviate the issue of the model easily getting lost in the middle. *e.g.*, CREAM-YaRN outperforms PoSE-YaRN [Zhu et al., 2023] by over 20% on average in the "Lost in the Middle" [Liu et al., 2024] task.
2. CREAM can further be enhanced by integrating novel designs on positional interpolation frequencies (such as Linear [Chen et al., 2023], NTK [Peng and Quesnelle, 2023], Yarn [Peng et al., 2023], *etc.*), and can be extended to context window sizes of up to 256K or beyond.
3. CREAM-Chat model requires only 100 steps of instruction-tuning to achieve nearly perfect performance on the Needle-in-a-Haystack pressure test, and it outperforms existing strong baselines on LongBench [Bai et al., 2023].

## 2   Methodology

### 2.1   Preliminaries

**Problem Formulation.**   Given an LLM with a pre-trained context window size $N$, our goal is to unlock the inference capacity of the LLM on the testing data $\mathcal{D}_{\text{test}}$ with an extended context window size $L$ (where $L > N$) by *efficiently* learning from a small-scale training data $\mathcal{D}_{\text{train}}$ with a maximum sequence length $N$. We expect the extended model to perform reasonably well in long-context evaluation.

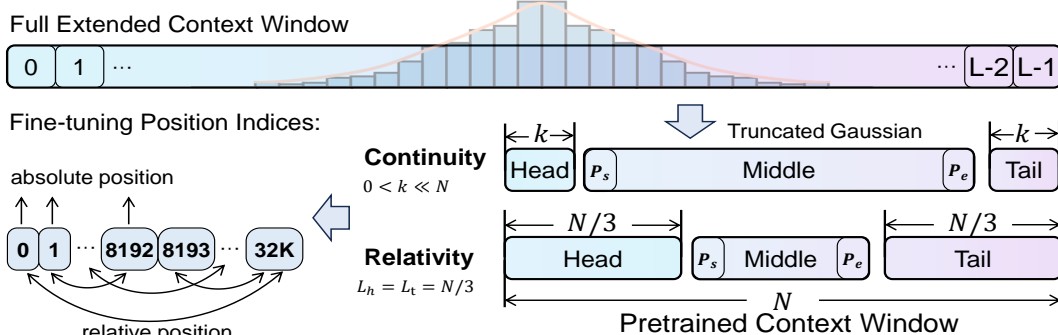

Figure 2: **Illustration of CREAM position interpolation.** The pre-trained context window is divided into three segments: the head, middle, and tail. To ensure continuity, we fix the lengths of the head and tail to a small value $k$. To maintain relativity, we set the lengths of the head and tail to $N/3$. For the middle part, the start and end position indices are determined via truncated Gaussian sampling, thereby encouraging the model to pay more attention to the information in the middle part.

**Continuity in Positional Encoding.** Transformer-based language models typically encode positional indices sequentially as $\{0, 1, \cdots, N-1\}$. Traditional length extension methods [Chen et al., 2023, Peng and Quesnelle, 2023, Peng et al., 2023] directly fine-tune on the target length $L$ with an updated positional index. This approach preserves the continuity of all absolute positions and learns all position indices within $[0, L-1]$, thereby successfully extending to the target length. Furthermore, PoSE [Zhu et al., 2023] attributed their superior performance over RandPos [Ruoss et al., 2023] to the ensured continuity of segments during fine-tuning.

**Relativity in Positional Encoding.** Relative positional encoding (RPE) [Shaw et al., 2018] has been proposed as an effective positional encoding method, where only the relative positions between two tokens are considered. Similar to prior works [Ruoss et al., 2023, Zhu et al., 2023, Wu et al., 2024], our work focuses on rotary positional encoding (RoPE) [Su et al., 2024], which is one of the most prominent RPE methods and has been widely applied to LLMs including the recent Llama family [Touvron et al., 2023b,a, AI@Meta, 2024]. In RoPE, only the relative distances between position pairs $(|j - i|; 0 \le i < j \le L - 1)$ are learned during fine-tuning (Appendix A). Due to this property, we can manipulate the position indices such that all relative positions between $[0, L-1]$ are learnable within the pre-trained window size.

## 2.2 Proposed Recipe: Continuity-Relativity indExing with gAussian Middle (CREAM)

In the following section, we start by introducing our design of dividing the context window $N$ to learn relative positional information. Then, we propose two strategies that target continuity and relativity, respectively. Lastly, we propose a novel truncated Gaussian sampling method to enhance the middle part of the long context. The overall framework is depicted in Figure 2.

**Context division.** We first discuss the motivations behind our design of the context length. First, prior works [Han et al., 2023, Xiao et al., 2023] observed that a significant amount of attention score is allocated to the beginning tokens of a sequence, which can potentially encode absolute positional information even without explicit positional encoding [Kazemnejad et al., 2024]. Secondly, the starting and ending tokens of long contexts can be treated as two pointers that localize the middle indices with the help of relative encodings. Therefore, we divide the pre-trained context window into three segments. The detailed ablation results are shown in Section 3.6.

**Definition 2.1.** Given the pre-trained context window size $N$ and target extended length $L$, the position set of $\{Head, Middle, Tail\}$ is defined as follows:

$$
\begin{aligned}
\text{Head} &= \{0, 1, ..., L_h - 1\}, \\
\text{Middle} &= \{P_s, P_s + 1, ..., P_e - 1, P_e\}, \\
\text{Tail} &= \{L - L_t, ..., L - 2, L - 1\}, \\
s.t. \; L_h &+ (P_e - P_s) + L_t = N,
\end{aligned}
\tag{1}
$$

where $L_h$ and $L_t$ denote the length of the head and tail segments, $P_s$ and $P_e$ denote the start and end position index of the middle segment.

The relative positions among the three segments in each sample are calculated in pairs, *i.e.*, $\{|j - i|; \forall i, j \in \{Head, Middle, Tail\}\}$.

The formed relative distance union $D_r$ learned by the model is given by:

$$[0, \max(L_h - 1, P_e - P_s, L_t - 1)] \cup [P_s - L_h + 1, P_e] \cup [L - L_t - P_e, L - 1 - P_s] \cup [L - L_t - L_h + 1, L - 1]. \quad (2)$$

Given that not all samples possess the same values for $L_h$, $P_s$, $P_e$, and $L_t$, as fine-tuning progresses, the union $D_r$ in Equation (2) can encompass the entire range $[0, L - 1]$, facilitating the model to learn all relative positions within the target length $L$.

**Two segmentation strategies.** For the sake of **continuity**, we set the $L_h$ and $L_t$ to a very small value $k$, where $0 < k \ll N$. Specifically, we use $k = 32$ in our experiments. This choice allows the middle segment to closely approximate the pre-trained context window. To maintain **relativity**, we divide $N$ equally into three parts and fix the $L_h$ and $L_t$ to $N/3$, enabling the model to learn as many relative positions as possible. In our fine-tuning process, both types of examples are sampled with equal probability to maintain balance.

**Truncated Gaussian Middle Sampling** To better focus the training process on the middle part of the long context, we introduce a truncated Gaussian function. This approach reduces the interval overlap in Equation (2) and directs the model's attention toward the middle section of the long context. In Appendix B, we provide theoretical justifications of our truncated Gaussian design, indicating that the maximization of $|D_r|$ holds for middle positions in $[N, L/2) \cup (L/2, L - N]$.

Formally, given the probability density function (PDF) of a Gaussian distribution:

$$f(x) = \frac{1}{\sigma\sqrt{2\pi}} \exp\left(-\frac{(x - \mu)^2}{2\sigma^2}\right),$$

where $\mu$ is the mean and $\sigma$ is the standard deviation. The corresponding cumulative distribution function (CDF) is:

$$F(x) = \int_{-\infty}^{x} f(t)\, dt = 0.5\left(1 + E\left(\frac{x - \mu}{\sigma\sqrt{2}}\right)\right), \quad E(z) = \frac{2}{\sqrt{\pi}} \int_0^z e^{-t^2}\, dt, \quad (3)$$

where $E(\cdot)$ is the error function. To calculate the CDF value within the truncated interval, we use a sufficiently large number (*e.g.* 1000) of equally spaced $x$ values from the given interval $[1, L/N]$:

$$x_i = 1 + \frac{(1 \times (L/N)) \cdot (i - 1)}{999}, \quad i = 1, 2, \ldots, 1000, \quad (4)$$

By substituting Equation (4) into Equation (3), the cumulative distribution function (CDF) curve is derived within the truncated interval. For sampling from this truncated Gaussian distribution, the inverse transform method is employed, as demonstrated in Equation (5):

$$\alpha = \text{round}(x_{i-1} + \frac{(x_i - x_{i-1})(u - F(x_{i-1}))}{F(x_i) - F(x_{i-1})}), \quad (5)$$

where $u \sim \text{Uniform}(0, 1)$, round$(\cdot)$ represents rounding to the nearest integer. Finally, we can get:

$$P_e \sim \text{Uniform}(L_h + \alpha \times L_m, (\alpha \times N - 1) - L_t), \\ P_s = P_e - L_m + 1, \quad (6)$$

where $L_m$ denotes the length of the middle segments. In summary, the overall sampling flow of our algorithm is presented in Algorithm 1.

## 3 Experiments

### 3.1 Experimental Setup

**Extended Models** We use `Llama-2-7B` and `Llama-2-7B-Chat` [Touvron et al., 2023a] as the base models and extend their pre-trained context window size of 4K to a target context length of 32K. The extended models are referred to as `CREAM-Base` and `CREAM-Chat`, respectively. Note that, though the target context length is 32K, we do not have to fine-tune `CREAM` on 32K token long text (see Section 2.2).

---

**Algorithm 1** `CREAM` sampling algorithm

---

**Require:** Pre-trained context window size $N$, extended context window size $L$, training sample size $S$, mean $\mu$, variance $\sigma$ and hyperparameter $k$.
1: Generate enough $x$ equally spaced according to Equation (4).
2: Substitute $x$ into Equation (3) to derive the truncated Gaussian CDF $F(x)$.
3: **for** $i = 0$ to $S - 1$ **do**
4:     Sample $L_h \sim \text{DiscreteUniform}(\{k, N/3\})$, and let $L_t = L_h, L_m = N - L_h - L_t$.
5:     Sample $u \sim \text{Uniform}(0, 1)$, and substitute it into Equation (5) to get $\alpha$.
6:     Calculate the start and end position ids $P_s, P_e$ of the middle part according to Equation (6).
7:     Get position set $P_i = \{0, 1, \ldots, L_h, P_s, \ldots, P_e, L - L_t \ldots, L - 1\}$, where $|P| = N$.
8: **end for**
9: **return** $P = \{P_0, P_1, \ldots, P_{S-2}, P_{S-1}\}$.

---

**Benchmarks**    We conduct long-context LLM evaluation of `CREAM`-Base on LongChat-Lines [Pal et al., 2023] and Lost-in-the-Middle [Liu et al., 2024]. Ideally, fine-tuning should not disrupt what the base model has learned, so we further evaluate `CREAM`-Base on the language modeling task and the evaluation benchmark [Beeching et al., 2023] adopted by Llama2. Additionally, we assess the `CREAM`-Chat model with Needle-in-a-Haystack[1] and LongBench[Bai et al., 2023]. Unless otherwise specified, we use linear interpolation to adapt LLMs to a longer context length.

**Baselines**    As far as we know, RandPos [Ruoss et al., 2023] and PoSE [Zhu et al., 2023] are similar to our approach in that they manipulate position indices to enable fine-tuning on the pre-trained length for context expansion. Therefore, these two methods serve as the baselines for our primary comparisons. More details about the experimental setup can be found in the Appendix C.

### 3.2   Effective Context Window Size Evaluation on `CREAM`-Base

We evaluate the long-context understanding capabilities of the `CREAM`-Base model on two tasks: LongChat-Lines[2] [Pal et al., 2023] (Figure 3) and "Lost in the Middle" [Liu et al., 2024] (Table 1).

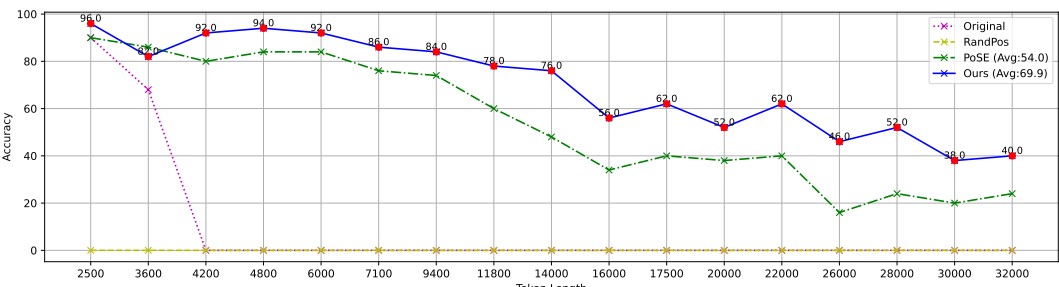

Figure 3: **Results (%) on LongChat-Lines**. Each length consists of 50 samples. All results are fine-tuned on `Llama-2-7B` with 4K length data through linear position interpolation. Refer to Appendix E for ablated results using NTK [Peng and Quesnelle, 2023] and Yarn [Peng et al., 2023].

**`CREAM`-Base performs best in retrieving information from long contexts of varying lengths.**    We extend the context window size up to 32K and compare `CREAM` with the `Llama 2-7B` [Touvron et al., 2023a], RandPos [Ruoss et al., 2023], and PoSE [Zhu et al., 2023]. As the context window size increases, the performance of all models drops, but `CREAM` always performs best except for the window size of 3.6K (see Figure 3). In terms of the average performance over all context window sizes, `CREAM` outperforms PoSE by 16%, demonstrating its good long-context understanding ability.

**`CREAM`-Base alleviates the Lost-in-the-Middle issue.**    Lost-in-the-Middle is an observation that LLMs are generally good at retrieving relevant information appearing at the beginning/end of the input context [Liu et al., 2024]. To validate the effectiveness of our middle-focused truncated Gaussian

---

[1]https://github.com/gkamradt/LLMTest_NeedleInAHaystack
[2]Passkey retrieval [Mohtashami and Jaggi, 2023] is another similar task for evaluating long-context LLMs, but it is too simplistic to reflect model performance at different context window sizes, so we use the dataset provided by Pal et al. [2023], which closely aligns with the task described in Li et al. [2023b]

Table 1: **Results (%) on "Lost in the Middle"**. "Position" indicates the correct answers' index, and each index comprises 500 samples. All results are fine-tuned on `Llama-2-7B` with 4K length data.

| Model | Position (75 keys, ∼5K tokens) | | | | | AVG | Position (140 keys, ∼10K tokens) | | | | | AVG |
|---|---|---|---|---|---|---|---|---|---|---|---|---|
| | **0** | **18** | **37** | **54** | **74** | | **0** | **34** | **69** | **104** | **139** | |
| PoSE-Linear | 99.4 | 24.4 | 37.4 | 47.2 | 46.2 | 50.9 | 95.2 | 8.2 | 7.6 | 13.8 | 18.6 | 28.7 |
| CREAM-Linear | 99.6 | 45.6 | 56.0 | 67.0 | 58.0 | **65.2** | 96.6 | 19.8 | 23.4 | 31.0 | 26.2 | **39.4** |
| PoSE-NTK | 98.6 | 49.6 | 44.6 | 40.2 | 41.4 | 54.9 | 97.6 | 3.4 | 0 | 0 | 27.6 | 25.7 |
| CREAM-NTK | 96.2 | 53.8 | 52.6 | 72.8 | 42.0 | **63.5** | 78.6 | 5.2 | 6.0 | 23.4 | 41.8 | **29.9** |
| PoSE-YaRN | 99.6 | 32.6 | 12.2 | 57.2 | 48.4 | 50.0 | 91.8 | 0.6 | 2.8 | 8.2 | 18.8 | 24.4 |
| CREAM-YaRN | 100.0 | 49.6 | 47.6 | 77.4 | 92.6 | **73.4** | 99.4 | 8.0 | 5.8 | 43.8 | 69.2 | **45.2** |

sampling, we evaluate `CREAM` and compare it with PoSE on the key-value retrieval task proposed by Liu et al. [2024]. We present results in Table 1, where the cyan shading indicates middle segments. We find that: regardless of the chosen interpolation method, `CREAM` always outperforms PoSE by a large margin. *e.g.*, `CREAM`-Linear surpasses PoSE-Linear by 21.2% when the relevant information is placed at 18.

## 3.3 Long Context Understanding Evaluation on `CREAM`-Chat

We conduct long-context evaluations of `CREAM`-Chat on two tasks:

- **Needle in A Haystack** (Figure 10)    This task is a test that places an answer (*i.e.*, Needle) at any position of a long context window (*i.e.*, Haystack) and requires a model to retrieve the correct answer given a question-answer pair. We follow Wu et al. [2024] and use the GPT (`GPT-3.5-Turbo-0125`) score as the evaluation metric.
- **LongBench** (Table 2)    Bai et al. [2023] is a more realistic benchmark because it covers real-world application scenarios like long-context QA and summarization. Moreover, it is specifically designed for Chat models.

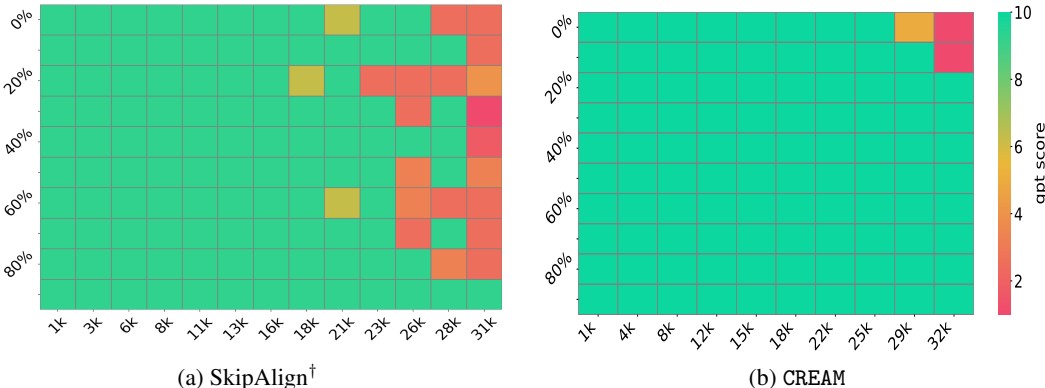

(a) SkipAlign[†]

(b) `CREAM`

Figure 4: **Results on Needle-in-a-Haystack.** [†] indicates the results excerpted from Wu et al. [2024]. Both results are instruction-tuned on `LLaMa2-7B-Chat` with 4K length data. The color gradually changes from deep green to deep red, indicating the Recall performance decreases from 10 to 1.

**`CREAM`-Chat outperforms SkipAlign in context window expansion.**    We visualize the results of `CREAM`-Chat and the recent SkipAlign in Figure 10. Clearly, `CREAM`-Chat beats SkipAlign because the performance of SkipAlign [Wu et al., 2024] decreases from the window size of 18K while `CREAM`-Chat displays a perfect performance everywhere until from the window size of 29K. Notably, `CREAM`-Chat is only fine-tuned for 100 steps.

**`CREAM`-Chat makes best use of the extended context window size.**    We present results on Long-Bench in Table 2. `CREAM`-Chat again surpasses strong baseline models, demonstrating its better use of extended context size. In terms of the average performance over all tasks, it outperforms the second

Table 2: **Results (%) on LongBench**. * indicates results reported by Bai et al. [2023]. CREAM-7B-32k is instruction-tuned for 100 steps using 4K length data on `LLaMa2-7B-Chat`.

| Model | Single-Doc QA | Multi-Doc QA | Summari-zation | Few-shot Learning | Code Completion | Synthetic Tasks | AVG |
|---|---|---|---|---|---|---|---|
| Llama2-7B-chat-4k* | 24.9 | 22.6 | 24.7 | 60.0 | 48.1 | 5.9 | 31.0 |
| XGen-7B-8k* | 24.6 | 20.4 | 24.7 | 56.2 | 38.6 | 5.3 | 28.3 |
| Mistral-7B-Instruct-v0.1 | 29.5 | 20.7 | 26.4 | 13.6 | 29.6 | 10.8 | 21.8 |
| Mistral-7B-Instruct-v0.2 | 28.5 | 21.5 | 26.1 | 50.1 | 33.8 | 13.9 | 29.0 |
| Mistral-7B-Instruct-v0.3 | 33.2 | 30.6 | 26.8 | 56.4 | 15.3 | 10.4 | 28.8 |
| InternLM-7B-8k* | 17.4 | 20.2 | 16.1 | 50.3 | 36.4 | 4.5 | 24.2 |
| Vicuna-v1.5-7B-16k* | 28.0 | 18.6 | 26.0 | 66.2 | 47.3 | 5.5 | 31.9 |
| LongChat-v1.5-7B-32k* | 28.7 | 20.6 | 26.7 | 60.0 | 54.1 | 15.8 | 34.3 |
| CREAM-7B-32k | 34.8 | 31.1 | 27.2 | 65.1 | 50.4 | 7.0 | **35.9** |

best model, *i.e.*, `LongChat-v1.5-7B-32k` [Li et al., 2023b], by 1.6%, though it is only tuned on a very small amount of data and for only 100 steps.

## 3.4 Effectiveness of PEFT Integration

To demonstrate that `CREAM` can be directly combined with PEFT techniques (such as LoRA [Hu et al., 2022] and QLoRA [Dettmers et al., 2023]), requiring no additional modifications. We conducted experiments on `LLaMa-2-7B-Chat` using the identical dataset and settings. The experimental results are presented in Table 3. The results indicate that models fine-tuned using LoRA and QLoRA achieve performance nearly equivalent to those fine-tuned with full parameter.

Table 3: **Results (%) on LongBench**. * indicates results reported by Bai et al. [2023]. CREAM-7B-32k is instruction-tuned for 100 steps using 4K length data on `LLaMa2-7B-Chat`.

| Model | Single-Doc QA | Multi-Doc QA | Summari-zation | Few-shot Learning | Code Completion | Synthetic Tasks | Macro |
|---|---|---|---|---|---|---|---|
| Llama2-7B* | 24.9 | 22.6 | 24.7 | 60.0 | 48.1 | 5.9 | 31.0 |
| LoRA | 28.9 | 28.6 | 27.8 | 62.2 | 54.6 | 10.8 | 35.5 |
| QLoRA | 28.1 | 27.6 | 28.1 | 61.7 | 54.6 | 10.1 | 35.0 |
| CREAM-7B-32k | 34.8 | 31.1 | 27.2 | 65.1 | 50.4 | 7.0 | 35.9 |

## 3.5 Language Modeling and Standard Benchmark

Following Chen et al. [2023], Zhu et al. [2023], Peng et al. [2023], we perform the classic language modeling evaluation, *i.e.*, perplexity evaluation, on GovReport [Huang et al., 2021] and Proof-pile [Zhangir Azerbayev, 2022]. Since a lower perplexity does not necessarily imply better model performance on downstream tasks [Zhang et al., 2024, Hu et al., 2024, Arora et al., 2024, Park et al., 2024], we further conduct evaluation on the standard natural-language-understanding (NLU) benchmark [Beeching et al., 2023]. This also lets us know whether fine-tuning hurts the NLU ability of the pre-trained base model.

**Both `CREAM` and PoSE demonstrate the lowest perplexity.** We apply different positional inter-polation methods to RandPos [Ruoss et al., 2023], PoSE [Zhu et al., 2023], and `CREAM` and report their perplexities in Table 4. We find that: `CREAM` and PoSE have a similar perplexity in different settings and both outperform RandPos. This occurs primarily because the position indices used during RandPos fine-tuning are discontinuous, which creates an inconsistency with the pre-training stage.

**`CREAM` has nearly the same NLU abilities as the pre-trained base model.** Ideally, fine-tuning should not adversely affect the original capabilities of the pre-trained base model. Our evaluation of `CREAM` confirms this, *i.e.*, `CREAM` nearly retains all NLU abilities of the base Llama2-7B (see Table 5). Interestingly, `CREAM` improves over Llama2-7B on ARC-C and HellaSwag. This is because

Table 4: **Perplexity results of GovReport and Proof-pile.** Each experiment is the average perplexity of 50 samples, and all results are based on `LLaMa-7B` fine-tuned on 4K data length.

| Model | GovReport | | | | Proof-pile | | | |
|---|---|---|---|---|---|---|---|---|
| | **4K** | **8K** | **16K** | **32K** | **4K** | **8K** | **16K** | **32K** |
| Original | 3.6 | - | - | - | 4.6 | - | - | - |
| RandPos-Linear | 8.9 | 7.4 | 6.2 | 5.8 | 12.1 | 11.9 | 11.9 | 12.9 |
| PoSE-Linear | 3.8 | 3.2 | 2.7 | 2.5 | 4.7 | 4.6 | 4.6 | 4.4 |
| CREAM-Linear | 3.8 | 3.2 | 2.7 | 2.5 | 4.7 | 4.6 | 4.5 | 4.4 |
| RandPos-NTK | 4.6 | 4.0 | 3.6 | 4.0 | 5.8 | 5.8 | 6.2 | 7.3 |
| PoSE-NTK | 3.7 | 3.2 | 2.7 | 2.6 | 4.7 | 4.6 | 4.5 | 4.7 |
| CREAM-NTK | 3.8 | 3.2 | 2.7 | 2.7 | 4.7 | 4.6 | 4.5 | 4.7 |
| RandPos-YaRN | 5.0 | 4.4 | 4.0 | 4.6 | 6.4 | 6.5 | 6.8 | 9.1 |
| PoSE-YaRN | 3.7 | 3.2 | 2.7 | 2.5 | 4.6 | 4.6 | 4.5 | 4.4 |
| CREAM-YaRN | 3.7 | 3.2 | 2.7 | 2.5 | 4.6 | 4.6 | 4.5 | 4.4 |

Table 5: **Experimental results of standard benchmarks.** [*] indicates results cited from Touvron et al. [2023a], and all results are based on `LLaMa-7B` fine-tuned on 4K data length.

| Model | Zero-Shot | | | | Few-Shot | |
|---|---|---|---|---|---|---|
| | **WinoGrande** | **TruthfulQA(mc2)** | **PIQA** | **BoolQ** | **ARC-C** | **HellaSwag** |
| LLaMa-2-7b-hf[*] | 69.2 | 39.5 | 78.8 | 77.4 | 45.9 | 77.2 |
| RandPos-Linear | 63.3 | 39.3 | 76.5 | 66.6 | 32.0 | 48.5 |
| PoSE-Linear | 68.8 | 38.6 | 77.8 | 76.2 | 47.7 | 77.1 |
| CREAM-Linear | 67.5 | 37.4 | 78.5 | 75.4 | 46.8 | 76.9 |
| RandPos-NTK | 68.7 | 35.9 | 78.6 | 74.8 | 45.5 | 74.4 |
| PoSE-NTK | 68.8 | 38.6 | 77.8 | 76.2 | 47.7 | 77.1 |
| CREAM-NTK | 67.5 | 37.4 | 78.5 | 75.4 | 46.8 | 76.9 |
| RandPos-YaRN | 69.3 | 36.6 | 78.3 | 72.5 | 43.4 | 69.2 |
| PoSE-YaRN | 69.4 | 39.6 | 78.1 | 76.7 | 49.0 | 78.0 |
| CREAM-YaRN | 68.7 | 38.5 | 78.0 | 76.4 | 49.0 | 78.0 |

these two tasks are few-shot tasks with longer prompts, necessitating the assistance of long-context understanding.

**Extending the context length to 256K.** We push the limit and extend the context length of `Llama-2-7B` up to 256K. Following Zhu et al. [2023], we evaluate the extended model by calculating the average perplexity over 20 samples from PG-19 [Rae et al., 2019] and Book3 [Presser, 2020].[3] Since the PG-19 test set does have enough samples that are longer than 256K, we select a subset of samples from the PG-19 training set.

We experiment with target context lengths 64K, 96K, 128K, 192K, and 256K and apply different positional interpolation methods to the extended model (see Table 6). The results of PoSE [Zhu et al., 2023] in Table 6 are based on fine-tuning `LLaMa 1-7B` with 2K data length, and are provided for reference only. Surprisingly, the increase of the target context length brings little to no perplexity increase, demonstrating the stability of CREAM across different target context lengths, even when the target context is extremely long.

## 3.6 Ablation Study

To validate the effectiveness of our modeling choices, we further conduct an ablation study of three main components of CREAM: truncated Gaussian sampling, fixed start and end segments, and the trade-off between continuity and relativity.

**Truncated Gaussian sampling versus Uniform sampling.** We use truncated Gaussian sampling to encourage CREAM to make better use of the middle part of the context. As a comparison, we replace

---

[3]We use sliding window for calculation, with a window size of 32,768 and a sliding step size of 4,096.

Table 6: **Perplexity results of PG-19 and Book3.** * indicates results copied from Zhu et al. [2023], and CREAM is based on LLaMa2-7B fine-tuned on 4K data length.

| Model | PG-19 | | | | | Book3 | | | | |
|---|---|---|---|---|---|---|---|---|---|---|
| | 64K | 96K | 128K | 192K | 256K | 64K | 96K | 128K | 192K | 256K |
| PoSE-Linear-128K* | 22.47 | 26.77 | 31.18 | - | - | 43.62 | 57.08 | 70.87 | - | - |
| PoSE-NTK-128K* | 14.84 | 29.48 | 34.80 | - | - | 16.04 | 31.42 | 37.00 | - | - |
| PoSE-YaRN-128K* | 10.36 | 10.77 | 11.33 | - | - | 12.30 | 13.07 | 13.81 | - | - |
| CREAM-Linear-192K | 5.9 | 6.0 | 6.1 | 6.1 | - | 7.6 | 7.7 | 7.8 | 7.8 | - |
| CREAM-NTK-192K | 5.0 | 5.1 | 5.2 | 5.2 | - | 6.9 | 7.0 | 7.0 | 7.1 | - |
| CREAM-YaRN-192K | 5.0 | 5.2 | 5.2 | 5.3 | - | 7.0 | 7.1 | 7.1 | 7.1 | - |
| CREAM-Linear-256K | 7.8 | 8.0 | 8.0 | 8.1 | 8.2 | 10.2 | 10.3 | 10.5 | 10.7 | 10.8 |
| CREAM-NTK-256K | 5.1 | 5.3 | 5.3 | 5.4 | 5.4 | 7.2 | 7.3 | 7.3 | 7.3 | 7.4 |
| CREAM-YaRN-256K | 5.2 | 5.3 | 5.4 | 5.4 | 5.5 | 7.1 | 7.2 | 7.2 | 7.3 | 7.3 |

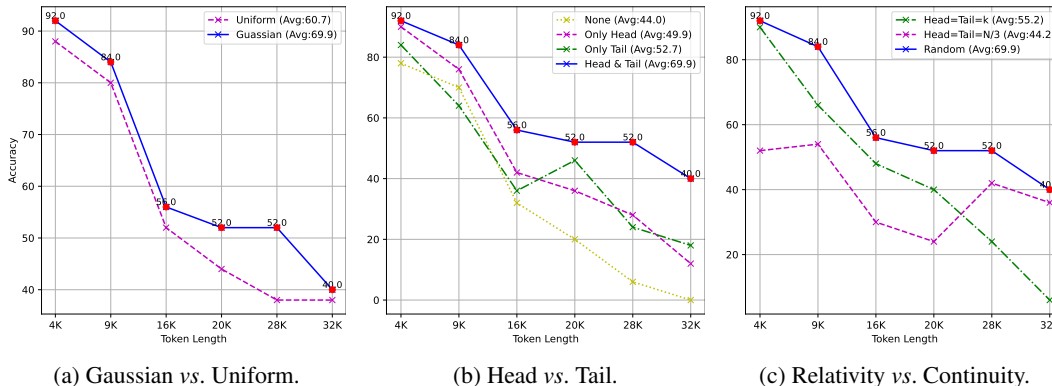

(a) Gaussian *vs*. Uniform.   (b) Head *vs*. Tail.   (c) Relativity *vs*. Continuity.

Figure 5: **Ablation study of CREAM on LongChat-Lines**. The result at each length is estimated using 50 samples.

it with the Uniform sampling (see Figure 5(a)). We observe that the Uniform sampling always leads to worse retrieval performance, suggesting the effectiveness of the truncated Gaussian sampling.

**Fixing the head and tail segments is crucial for good retrieval performance.** We compare our choice of fixing the head and tail segments with three alternatives: (i) removing both the head and tail segment, (ii) fixing only the head segment, and (iii) fixing only the tail segment (see Figure 5(b)). We find that: removing the head and tail segments leads to the worst performance; it results in a complete failure (*i.e.*, zero score) for the context size 32K. Keeping either head or tail segments performs slightly better than removing both but underperforms our default choice of fixing both. We suppose that this is because fixing both gives rise to better relativity information, a finding that is consistent with that of Han et al. [2023].

**Maintaining a good balance between continuity and relativity is necessary.** We encourage continuity by setting the head and tail segment lengths to $k = 32$ and elicit relativity by letting $k = N/3$ (see Section 2.2). To balance the two desired properties, we randomly choose $k = 32$ and $k = N/3$ with an equal probability during fine-tuning. Here we compare three scenarios: (1) enforce only continuity, (2) enforce only relativity, and (3) balance continuity and relativity (see Figure 5(c)). We find that balancing continuity and relativity gives rise to the best performance, thus justifying our modeling choice.

**Ablation of Hyperparameters** In our implementation of truncated Gaussian sampling, as illustrated in Equation (3), the only hyperparameters are the mean $\mu$ and the variance $\sigma$. The mean $\mu$ is determined by the expansion factor. The variance $\sigma$ is adaptable based on data, we conducted experiments with five different values of $\sigma$. The results, as presented in Figure 6, indicate that the current selection ($\sigma = 3$) yields optimal performance.

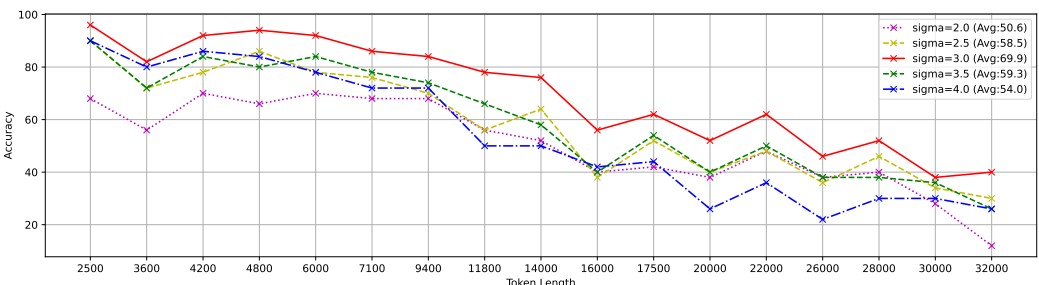

Figure 6: **Ablation Results (%) on LongChat-Lines**. Each length consisting of 50 samples. The above are the results of using Linear interpolation on the `Llama 2-7B` model.

## 4 Related Works

**Efficient Transformers and Extra Memory**    FoT [Tworkowski et al., 2024] addresses the limitations of local attention in transformers by integrating memory attention layers, which enable large models to learn from a wide context while reducing interference. Infini-attention [Munkhdalai et al., 2024] incorporates compressed memory into the standard attention mechanism and integrates masked local attention and long-term linear attention mechanisms within a single Transformer block. LLoCO [Tan et al., 2024] employs LoRA in conjunction with context compression, retrieval, and parameter-efficient fine-tuning to learn context offline. Although these methods can successfully extend the long context window of LLMs, they either require modifications to the attention mechanism or the addition of extra modules for assistance. In contrast, CREAM does not require these operations and can be directly applied to a pre-trained model.

**Positional Interpolation**    Chen et al. [2023] first proposed extending the context window through positional interpolation, which linearly reduces the input position indices to match the original context window size, thereby preventing catastrophic high attention scores from completely disrupting the self-attention mechanism. Subsequently, various methods (such as NTK [Peng and Quesnelle, 2023], ABF [Xiong et al., 2023], and EABF [Zhang et al., 2024]) emerged that modify the base frequency of rotary positional encoding to achieve positional interpolation. YaRN [Peng et al., 2023] introduced a segmented interpolation method, applying different positional interpolations to different dimensions. LongRoPE [Ding et al., 2024] identifies and utilizes two forms of non-uniformity in positional interpolation through search, and introduces a progressive expansion strategy for positiona interpolation. Moreover, CREAM can be combined with any positional interpolation method.

**Positional Encoding**    RandPos [Ruoss et al., 2023] first modified position indices so that the model leverages the relativity of positions, enabling it to extend to the target length with fine-tuning over shorter lengths. PoSE [Zhu et al., 2023] then emphasized the importance of continuous segments, dividing the training length into two parts to further enhance the interpolation effect. CREAM utilizes both relativity and continuity, and it also better enables the model to focus on the middle part of the context.

## 5 Conclusion

We proposed **C**ontinuity-**R**elativity ind**E**xing with g**A**ussian **M**iddle (CREAM), a simple yet effective method to extend the context of large language models. CREAM achieves a trade-off between continuity and relativity, enabling the model to exploit positional relativity (*i.e.*, fine-tuning within the pre-trained length), while preserving text continuity (*i.e.*, remaining as close as possible to the pre-trained state). Furthermore, by employing truncated Gaussian sampling, the model can concentrate more on the middle positions during fine-tuning. Experimental results demonstrate that CREAM outperforms other methods on both Base and Chat models and effectively mitigates the issue of "lost in the middle".

## Acknowledgement

The authors thank the reviewers for their insightful suggestions to improve the manuscript. This work presented herein is supported by the National Natural Science Foundation of China (62376031).

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

## A Relative Positional Encoding in RoPE

We provide a simple background proof on the relative positional encoding performed by Rotary Position Embedding (RoPE) Su et al. [2024]. Given two embedding vectors $\boldsymbol{x}_q, \boldsymbol{x}_k \in \mathbb{R}^d$ corresponds to query and key at positions $(m, n) \in [0, L)$, where $d$ is embedding dimension, their encoding counterparts can be defined as:

$$\begin{aligned}
\boldsymbol{q}_m &= f_q(\boldsymbol{x}_q, m) = \mathbf{R}_{\Theta,m}^d(\boldsymbol{x}_q, m) \\
\boldsymbol{k}_n &= f_k(\boldsymbol{x}_k, n) = \mathbf{R}_{\Theta,n}^d(\boldsymbol{x}_k, n)
\end{aligned} \tag{7}$$

where

$$\mathbf{R}_{\Theta,m}^d = \begin{bmatrix}
\cos m\theta_1 & -\sin m\theta_1 & \cdots & 0 & 0 \\
\sin m\theta_1 & \cos m\theta_1 & \cdots & 0 & 0 \\
\vdots & \vdots & \ddots & \vdots & \vdots \\
0 & 0 & \cdots & \cos m\theta_{d/2} & -\sin m\theta_{d/2} \\
0 & 0 & \cdots & \sin m\theta_{d/2} & \cos m\theta_{d/2}
\end{bmatrix} \tag{8}$$

is the rotary matrix, $\Theta = \{\theta_i = 10000^{-2(i-1)/d}, i = [1, 2, \ldots, d/2]\}$ is pre-defined rotation angles. Then the self attention score can be obtained with:

$$\begin{aligned}
\boldsymbol{q}_m^{\mathrm{T}} \boldsymbol{k}_n &= \langle f_q(\boldsymbol{x}_q, m), f_k(\boldsymbol{x}_k, n) \rangle \\
&= \mathrm{Re} \left[ \sum_{i=0}^{d/2-1} \boldsymbol{x}_{q[2i:2i+1]} \boldsymbol{x}_{k[2i:2i+1]}^* e^{i(m-n)\theta_i} \right] \\
&:= g(\boldsymbol{x}_m, \boldsymbol{x}_n, m-n)
\end{aligned} \tag{9}$$

where $\boldsymbol{x}^*$ represents the conjugate complex of $\boldsymbol{x}$, $g$ is the derived attention function of RoPE. As seen, RoPE only depends on the relative distances between and encodes the relative position information.

## B Theoretical findings of `CREAM` design

**Theorem B.1.** *If $N \ll L$, the spanning size $|D_r|$ of the relative position union in Equation (2) reaches its maximum iff. one of the following groups of inequalities satisfies:*

$$\max(L_h - 1, P_e - P_s, L_t - 1) + L_h - 1 < P_s < P_e < (L - L_t)/2, \tag{10}$$

*or*

$$(L + L_h)/2 - 1 < P_s < P_e < L - L_t - \max(L_h - 1, P_e - P_s, L_t - 1), \tag{11}$$

*where* $\max |D_r| = \max(L_h - 1, P_e - P_s, L_t - 1) + 2N$.

*Proof.* Denote four intervals in Equation (2) as $S_i, i = 1, \ldots, 4$. According to the inequality of inclusion-exclusion principle for the cardinality of the union of $n$ sets:

$$|D_r| = |\cup_{i=1}^4 S_i| \le \sum_{i=1}^4 |S_i|, \tag{12}$$

where the equality holds *iff.* all sets are pairwise disjoint. That is

$$S_i \cap S_j = \varnothing, \quad \forall i \ne j \tag{13}$$

Given intervals as in Equation (2), we have

$$\begin{cases}
\mathrm{MAX} < P_s - L_h + 1 \\
P_e < L - L_t - P_e \\
L - 1 - P_s < L - L_t - L_h + 1
\end{cases} \quad or \quad \begin{cases}
\mathrm{MAX} < L - L_t - P_e \\
L - 1 - P_s < P_s - L_h + 1 \\
P_e < L - L_t - L_h + 1
\end{cases}, \tag{14}$$

where $\mathrm{MAX} = \max(L_h - 1, P_e - P_s, L_t - 1)$. The above inequalities can be simplified to Equations (10) and (11).

**Lemma B.2.** *Under mild assumptions that $L - L_t \approx L$, $L + L_h \approx L$, the maximization in Theorem B.1 holds for all $(P_s, P_e) \in [N, L/2) \cup (L/2, L - N]$.*

*Proof.* Given that

$$\begin{aligned}
\max(L_h - 1, P_e - P_s, L_t - 1) + L_h - 1 < \max(2L_h, N - L_t, N - L_m) < N \\
L - L_t - \max(L_h - 1, P_e - P_s, L_t - 1) > L - \max(N - L_m, N - L_h, 2L_t) > L - N,
\end{aligned} \quad (15)$$

the inequalities in Equations (10) and (11) turns into $[N, L/2) \cup (L/2, L - N]$.

**Theorem B.3.** *If $N \ll L$, when the spanning size $|D_r|$ of the relative position union in Equation (2) reaches its maximum, we denote the coverage area of the middle segment as:*

$$S_m := \left\{ x \middle| x \in [P_s, P_e], (P_s, P_e) \in \left\{ \arg\max_{(P_s, P_e)} |D_r| \right\} \right\} \quad (16)$$

*thus, we have:*

$$L \geq S_m + L_h + L_t > L - N/2 \quad (17)$$

*Furthermore, as $\frac{N}{L} \to 0$, we have:*

$$L_h + S_m + L_t \to L \quad (18)$$

## C  Experimental Details

**Model Hyperparameters**   We fine-tune all models by optimizing the causal language modeling objective. A learning rate of $2 \times 10^{-5}$ with a linear scheduler is adopted, incorporating 10 warm-up steps. We use the AdamW Loshchilov and Hutter [2018] optimizer with the hyperparameter configurations specified by PyTorch Paszke et al. [2019]. To speed up fine-tuning, we resort to DeepSpeed [4] ZeRO stage 1 and Flash Attention-2 Dao [2023]. We perform fine-tuning on two A100-80G GPUs with a total batch size of 32 and run inference on a single A100-80G GPU. For CREAM-Base, we fine-tune it for 1,000 steps on a dataset derived from Pile Gao et al. [2020]; for CREAM-Chat, we fine-tune it for 100 steps on ShareGPT Zheng et al. [2024]. To ensure fair comparison, we follow the fine-tuning and inference configurations established by Zhu et al. [2023].

**Datasets and Training Cost**   For training the Base model, we directly utilize The Pile data provided by Zhu et al. [2023], and select samples with token lengths exceeding 4K. For training the Chat model, we filter the ShareGPT data from public datasets[5]. Specifically, we used the Vicuna prompt template to sequentially concatenate the ShareGPT data until each data point comprises at least 4K tokens. Then, we select 3.2K data points to train for 100 steps. Particularly, during the instruction tuning process, we mask the USER part and allow the model to calculate the loss only on the ASSISTANT part. We utilize two A100-80G machines with a global batch size of 32, fully utilizing the available memory. Running 1,000 steps for the Base model takes approximately 6 hours, while running 100 steps for the Chat model takes approximately 2 hours.

## D  Robustness Across LLMs

Our proposed method exhibits strong generalization capabilities and can be applied to other large language models (LLMs) without the need for parameter modification. To validate this, we conducted experiments on Baichuan2-7B, with the corresponding results presented in Table 7.

Furthermore, we fine-tuned LLaMa3-8B using a context window size of $4K$ tokens, with the experimental outcomes shown in Table 8.

The results in Tables 7 and 8 clearly demonstrate the transferability of our method to different models, underscoring its robustness. Of particular note is that despite LLaMa3-8B having a native context length of $8K$ tokens, fine-tuning on training data with a $4K$ context window yielded unexpectedly strong performance.

---

[4] https://github.com/microsoft/DeepSpeed

[5] https://huggingface.co/datasets/Aeala/ShareGPT_Vicuna_unfiltered

Table 7: **Perplexity results of GovReport and Proof-pile.** Each experiment is the average perplexity of 50 samples, and all results are based on `Baichuan2-7B` fine-tuned on 4K data length.

| Model | GovReport | | | | Proof-pile | | | |
|---|---|---|---|---|---|---|---|---|
| | 4K | 8K | 16K | 32K | 4K | 8K | 16K | 32K |
| Original | 3.3 | - | - | - | 5.8 | - | - | - |
| CREAM-Linear | 3.6 | 2.9 | 2.5 | 2.2 | 6.2 | 6.1 | 6.0 | 5.8 |

Table 8: **Results (%) on LongChat-Lines**. Each length consists of 50 samples. All results are fine-tuned on `Llama-3-8B` with **4K** length data through linear position interpolation.

| AVG Length | 2000 | 4000 | 7800 | 8800 | 9700 | 11000 | 12000 | 14000 | 17000 | 19000 | 24000 | 28000 | 32000 |
|---|---|---|---|---|---|---|---|---|---|---|---|---|---|
| CREAM-Linear | 0.98 | 1.00 | 0.96 | 0.94 | 0.86 | 0.92 | 0.92 | 0.92 | 0.86 | 0.84 | 0.70 | 0.60 | 0.48 |

# E  LongChat Lines Results

The interpolation methods using NTK and Yarn are presented in Figures 7 and 8. As can be seen, CREAM performs the same as the Linear method for interpolation, still outperforming other methods. The result of NTK at 26K-32K is zero, which is due to the inherent properties of NTK, a finding that is aligns with Zhu et al. [2023].

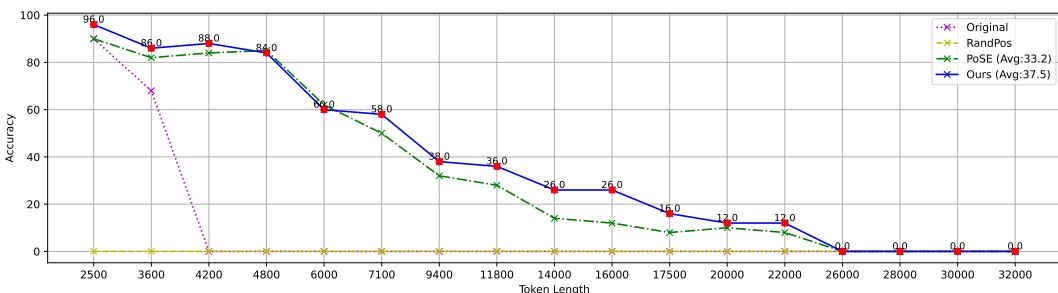

Figure 7: **Results (%) on LongChat-Lines**. Each length consisting of 50 samples. The above are the results of using NTK interpolation on the `Llama 2-7B` model.

# F  LongBench Subtasks Results

The results of each subtask in Tables 2 are shown in Tables 10 and 11.

It is noteworthy that, to provide further evidence of the efficacy of our model, we have specifically chosen 12 tasks from the four categories outlined in Zhang et al. [2024] for comparison purposes. As delineated in Table 9, we are able to attain superior performance on LongBench in comparison to EABF Zhang et al. [2024], even with shorter training lengths and less data.

# G  Limitations

When extending the context beyond the pre-trained length, there is an inevitable loss of information due to position interpolation, particularly when fine-tuning is restricted to the pre-trained length. However, in comparison to previous methods such as RandPos Ruoss et al. [2023] and PoSE Zhu et al. [2023], CREAM has effectively mitigated the issue of "Lost-in-the-Middle" by introducing truncated Gaussian sampling. Additionally, as discussed in reference Liu et al. [2024], decoder-only models are prone to inherently exhibiting a U-shaped performance curve on this task. Therefore, completely solving this problem remains challenging.

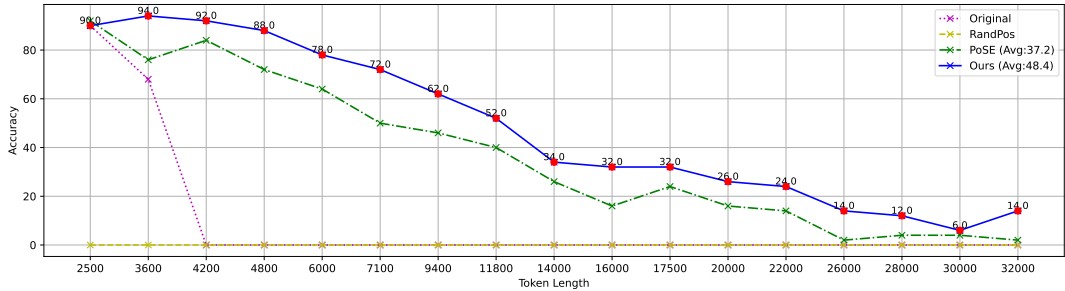

Figure 8: **Results (%) on LongChat-Lines**. Each length consisting of 50 samples. The above are the results of using Yarn interpolation on the `Llama 2-7B` model.

Table 9: **Experimental results (%) of the `LongBench` subtasks** selected in Zhang et al. [2024]. †
indicates results quoted from Zhang et al. [2024]. **Len** represents the context length during fine-tuning.
All results are based on `Llama 2-7B`.

| Model | Num / Len | Singl-Doc QA | | | Multi-Doc QA | | | Summarization | | | Few-shot Learning | | | AVG |
|---|---|---|---|---|---|---|---|---|---|---|---|---|---|---|
| | | NQA | QAPR | MFQA_en | HPQA | WMQA | MSQ | GR | QMSM | MNWS | TREC | TRVQA | SMSM | |
| PI† | | 20.1 | 30.4 | 45.3 | 26.1 | 30.1 | 9.9 | 28.1 | 23.7 | 26.6 | 68.0 | 84.9 | 42.5 | 36.3 |
| NTK-By-Parts† | | 15.9 | 31.1 | 40.1 | 25.4 | 26.6 | 7.2 | 26.7 | 22.4 | 26.9 | 68.5 | 82.8 | 42.9 | 34.7 |
| Yarn† | 3.5K / 16K | 20.3 | 28.9 | 42.8 | 27.8 | 30.7 | 7.2 | 27.4 | 22.5 | 26.8 | 66.0 | 85.6 | 42.6 | 35.7 |
| ABF† | | 24.6 | 32.8 | 45.6 | 35.1 | 30.3 | 15.2 | 30.8 | 23.0 | 27.4 | 71.0 | 84.7 | 42.7 | 38.6 |
| EABF† | | 21.9 | 31.0 | 47.1 | 40.1 | 32.7 | 15.1 | 32.3 | 23.0 | 27.1 | 70.5 | 86.7 | 42.0 | 39.1 |
| CREAM | **3.2K / 4K** | 23.0 | 34.6 | 46.8 | 42.2 | 33.7 | 17.4 | 30.4 | 24.3 | 26.8 | 69.5 | 84.0 | 41.9 | **39.6** |

Table 10: **Experimental results (%) of the LongBench subtasks.**

| Model | Singl-Doc QA | | | Multi-Doc QA | | | Summarization | | |
|---|---|---|---|---|---|---|---|---|---|
| | NQA | QAPR | MFQA_en | HPQA | WMQA | MSQ | GR | QMSM | MNWS |
| Llama2-7B-chat-4k* | 18.7 | 19.2 | 36.8 | 25.4 | 32.8 | 9.4 | 27.3 | 20.8 | 25.8 |
| XGen-7B-8k* | 18.0 | 18.1 | 37.7 | 29.7 | 21.1 | 10.3 | 27.3 | 20.5 | 26.2 |
| InternLM-7B-8k* | 12.1 | 16.7 | 23.4 | 28.7 | 22.8 | 9.0 | 9.7 | 15.9 | 22.8 |
| Vicuna-v1.5-7B-16k* | 19.4 | 26.1 | 38.5 | 25.3 | 20.8 | 9.8 | 27.9 | 22.8 | 27.2 |
| LongChat-v1.5-7B-32k* | 16.9 | 27.7 | 41.4 | 31.5 | 20.6 | 9.7 | 30.8 | 22.7 | 26.4 |
| CREAM | 23.0 | 34.6 | 46.8 | 42.2 | 33.7 | 17.4 | 30.4 | 24.3 | 26.8 |

Table 11: **Experimental results (%) of the LongBench subtasks.**

| Model | Few-shot Learning | | | Code Completion | | Synthetic Tasks | |
|---|---|---|---|---|---|---|---|
| | TREC | TRVQA | SMSM | PC | PR_en | LCC | RBP |
| Llama2-7B-chat-4k* | 61.5 | 77.8 | 40.7 | 2.1 | 9.8 | 52.4 | 43.8 |
| XGen-7B-8k* | 65.5 | 77.8 | 25.3 | 2.1 | 8.5 | 38.6 | 38.6 |
| InternLM-7B-8k* | 52.0 | 77.8 | 21.2 | 3.0 | 6.0 | 44.1 | 28.8 |
| Vicuna-v1.5-7B-16k* | 71.5 | 86.2 | 40.8 | 6.5 | 4.5 | 51.0 | 43.5 |
| LongChat-v1.5-7B-32k* | 63.5 | 82.3 | 34.2 | 1.0 | 30.5 | 53.0 | 55.3 |
| CREAM | 69.5 | 84.0 | 41.9 | 3.0 | 11.0 | 52.0 | 48.7 |

# H   Loss Curve

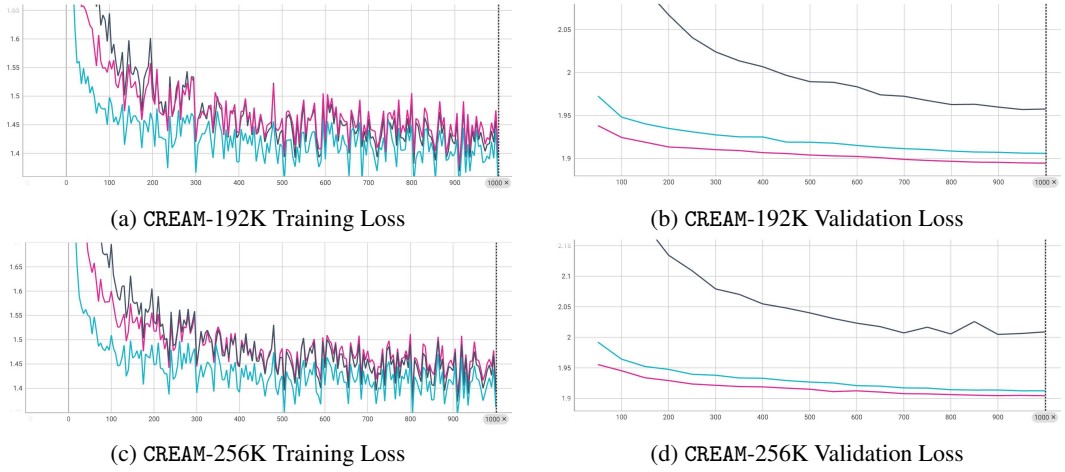

(a) CREAM-192K Training Loss      (b) CREAM-192K Validation Loss

(c) CREAM-256K Training Loss      (d) CREAM-256K Validation Loss

Figure 9: **Fine-tuning loss curve based on** `Llama 2-7B`**.** The black line represents Linear interpolation, the pink line represents NTK interpolation, and the cyan line represents YaRN interpolation.

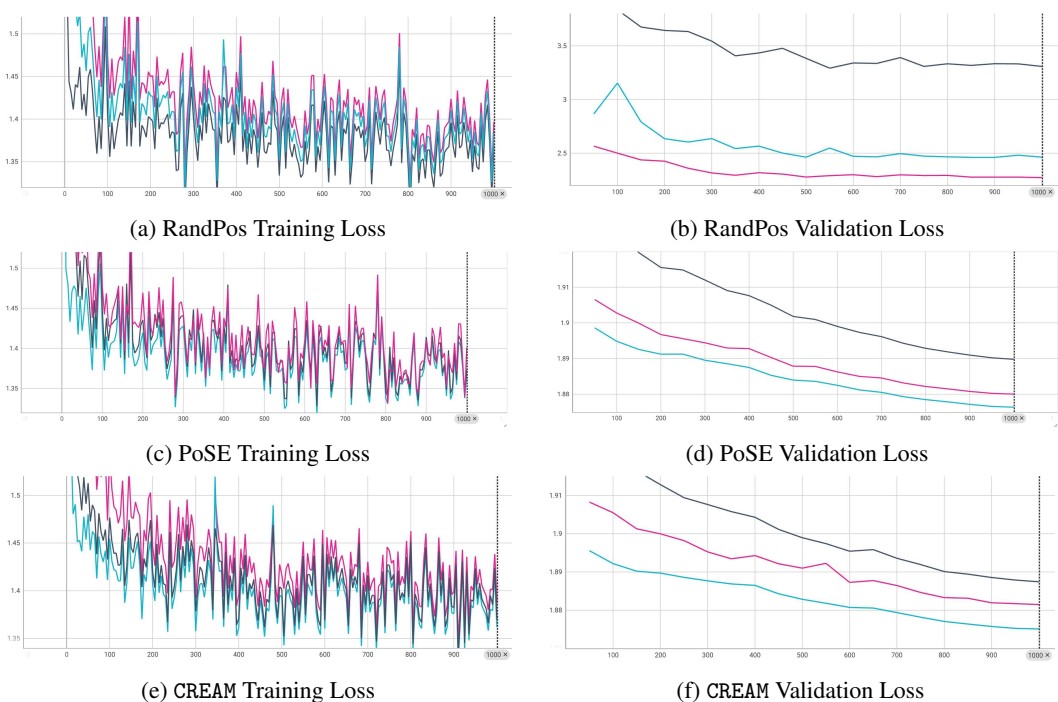

(a) RandPos Training Loss      (b) RandPos Validation Loss

(c) PoSE Training Loss      (d) PoSE Validation Loss

(e) CREAM Training Loss      (f) CREAM Validation Loss

Figure 10: **Fine-tuning loss curve based on** `Llama 2-7B`**.** The black line represents Linear interpolation, the pink line represents NTK interpolation, and the cyan line represents YaRN interpolation.

