# OpenReview forum: "An Efficient Recipe for Long Context Extension via Middle-Focused Positional Encoding"
_NeurIPS.cc/2024/Conference — NeurIPS 2024 poster_

### Official Review · Reviewer_qJ1n · 2024-06-27

**Soundness:** 2
**Presentation:** 1
**Contribution:** 2
**Rating:** 3
**Confidence:** 2

**Summary:**

The paper introduces CREAM (Continuity-Relativity indExing with gAussian Middle), an efficient method for extending the context window of large language models (LLMs) to handle longer contexts without the need for extensive fine-tuning. CREAM manipulates position indices for shorter sequences within the pre-trained context window, using two key strategies: continuity, and relativity. Additionally, CREAM employs a truncated Gaussian distribution to enhance the model's focus on the middle part of the context, mitigating the "Lost-in-the-Middle" problem. Comprehensive experiments demonstrate CREAM's efficiency and effectiveness, showing improved performance over baselines like RandPos and PoSE on various tasks.

**Strengths:**

CREAM achieves superior performance across a spectrum of long-context tasks. The empirical results substantiate CREAM's ability to effectively mitigate the "Lost-in-the-Middle" problem.

**Weaknesses:**

I think the presentation of this paper is a significant problem. Actually, I cannot understand the method itself nor the intuitions behind it. First, I do not understand the concept of continuity in positional encoding. The authors use a short paragraph to explain continuity (lines 69-74) without any formulation, which is very abstract. The same goes for relativity, though I understood it from another paper. After explaining these two concepts, you still need to explain why they are important (I did not see this in the paper). Second, I do not understand why splitting the pre-trained context into three segments with different sizes can achieve continuity and relativity. Third, I do not understand why truncated Gaussian middle sampling mitigates the "Lost-in-the-Middle" problem. Line 113 states that it reduces the interval overlap in Eq2. But what is interval overlap and why does it result in the "Lost-in-the-Middle" problem?

I may not be the ideal reader for this paper, but if other reviewers feel the same way, this paper may have been hastily completed.

**Questions:**

See Weaknesses

---

> ### Author Rebuttal · Authors · 2024-08-06
>
> Thank you very much for taking the time to review our paper. First and foremost, we would like to express our sincere apologies for any confusion that our work brings to you. As you may have noticed, our presentation reached an average of 3.33 / 4 among three other reviewers. Specifically, reviewer srX3 appreciates the clear writing in methods and fWsk confirms the excellent presentation. Nevertheless, we deeply respect your criticism and would like to further clarify all your confusion and misunderstandings **point-by-point** as follows.
>
> >**Q1:** First, I do not understand the concept of continuity in positional encoding. The authors use a short paragraph to explain continuity (lines 69-74) without any formulation, which is very abstract. The same goes for relativity, though I understood it from another paper. After explaining these two concepts, you still need to explain why they are important (I did not see this in the paper).
>
> **A1:** In short,  the concept of continuity in positional encoding lies in the importance of ensuring the consistency of position indices between fine-tuning and pre-training. Specifically,
>
> 1. In **lines 69-74,** we've discussed the continuity of positional encoding in detail  and provided relevant references. PoSE also highlights the importance of continuity in positional encoding during training.
> 2. In **lines 75-82**, we elaborate on the relativity of positional encoding and provide a theoretical proof in **Appendix B**.
> 3. In **lines 38-44**, we explain the roles of continuity and relativity. Our ablation experiments (Figure 5(c)) experimentally validate the importance of continuity and relativity.
>
> >**Q2:** Second, I do not understand why splitting the pre-trained context into three segments with different sizes can achieve continuity and relativity.
>
> **A2:** In lines 105-110 of our paper, we've provided a detailed explanation of this matter. The continuity design aims to ``allow the middle segment to closely approximate the pre-trained context window.`` The relativity design aims to ``enable the model to learn as many relative positions as possible.``
>
> >**Q3**: Third, I do not understand why truncated Gaussian middle sampling mitigates the "Lost-in-the-Middle" problem. Line 113 states that it reduces the interval overlap in Eq2. But what is interval overlap and why does it result in the "Lost-in-the-Middle" problem?
>
> **A3:** In **lines 111-115** of our paper, we note that truncated Gaussian sampling can alleviate the "lost in the middle" problem by ``directing the model's attention towards the middle section of the long context``. The overlap of intervals refers to the repeated position indices in Equation (2). The "lost in the middle" issue **is not caused by interval overlap but by a bias inherent in the model, as noted in [1]**. In Appendix B and L114-115, we further provide theoretical justification that sampling the middle part of a long context with a high importance rate yields a maximization of learned relative position intervals as identified in Equations (2).
>
>
> This work indeed has gone through extensive experiments, careful writing, and figure polishing. We tried to make all contents illustrative and provided detailed supplementary materials in appendices. We hope our answers together with appendices have provided sufficient explanations and clarified your confusion. We are more than willing to elaborate more details if you have further questions.
>
> Reference:
>
> [1] Lost in the middle: How language models use long contexts. TACL, 2024.

---

### Official Review · Reviewer_XDBh · 2024-07-09

**Soundness:** 3
**Presentation:** 3
**Contribution:** 3
**Rating:** 5
**Confidence:** 4

**Summary:**

The paper presents CREAM (Continuity-Relativity indExing with gAussian Middle), an innovative approach to extend the context window of Large Language Models (LLMs) without the need for extensive fine-tuning at the target length. The authors address the "Lost in the Middle" problem, which plagues long-context LLMs by causing a performance drop when retrieving information from the middle of the context. CREAM achieves this by manipulating position indices and introducing a truncated Gaussian sampling method to focus on the middle part of the context during fine-tuning.

**Strengths:**

1. **Novel Approach**: CREAM offers a novel positional encoding strategy that efficiently extends the context window of LLMs, which is a significant contribution to the field.

2. **Empirical Evidence**: The paper provides strong empirical evidence through comprehensive experiments, demonstrating CREAM's superiority over existing methods like PoSE and RandPos, especially in handling middle context information.

3. **Training Efficiency**: The method requires fine-tuning at the pre-trained context window size, which is computationally efficient compared to fine-tuning at the target length.

4. **Theoretical Foundation**: The paper includes theoretical justifications for the use of truncated Gaussian distribution, adding rigor to the proposed method.

**Weaknesses:**

1. **Generalizability**: While the paper shows impressive results, it is not clear how generalizable these findings are to other LLMs beyond the tested Llama2-7B model. And I'm wondering whether the method can work with PEFT tuning techniques, such as QLoRA.

2. **Lack of Comparative Analysis**: The paper could benefit from a more detailed comparative analysis with other contemporary methods. For example, in Table 2, the paper only compares one model (LongChat-v1.5-7B-32k) with the same 32k context window. I suggest that in such comparision, peer models should at least take the same length of input, otherwise, it cannot prove the effectiveness of the proposed model over other Long LLMs with 32k+ context windows. I suggest the author compare with models like Mistral-7B-Instruct-v0.2, LongLoRA etc.

**Questions:**

1. What are the computational overheads associated with implementing CREAM, and how does it scale with larger context sizes?
2. Are there any specific hyperparameter settings in CREAM that are particularly sensitive, and how were these chosen?

**Limitations:**

The discussion in Appendix F regarding the limitations of the CREAM method could be misleading. The author implies that CREAM provides enhanced performance compared to other methods concerning the "Lost in the Middle" issue but acknowledges that the problem is not fully solved due to the nature of decoder-only models. For a genuine limitations section, it is essential to delve deeper into the specific constraints of the CREAM approach and how these limitations might impact its application and effectiveness. A more explicit acknowledgment of the method's shortcomings and a clear explanation of why these issues cannot be overcome with the current model design would significantly improve the section's clarity and usefulness.

---

> ### Author Rebuttal · Authors · 2024-08-06
>
> Thank you very much for taking the time to review our paper and acknowledging the novelty and empirical superiority of our approach. Below, we provide detailed replies to your comments and hope we can resolve your major concerns.
>
> >**W1:** **Generalizability**: While the paper shows impressive results, it is not clear how generalizable these findings are to other LLMs beyond the tested Llama2-7B model. And I'm wondering whether the method can work with PEFT tuning techniques, such as QLoRA.
>
> **A1:** Thank you for the constructive comments. Our method has strong generalization capabilities and **can be applied to other LLMs without modifying any parameters**. To verify this, we conducted experiments on Baichuan2-7B. The experimental results are shown in the table below. (Incidentally, as shown in the table in A2 of reviewer srX3's response, the experimental results on LLaMa3-8B also support this conclusion.)
>
> |            | GovReport |      |      |      | Proof-pile |      |      |      |
> | :---------------- | :-------- | :--- | :--- | :--- | :--------- | :--- | :--- | :--- |
> |                   | 4K        | 8K   | 16K  | 32K  | 4K         | 8K   | 16K  | 32K  |
> | Baichuan2-7B-Base | 3.3       | -    | -    | -    | 5.8        | -    | -    | -    |
> | CREAM-Linear      | 3.6       | 2.9  | 2.5  | 2.2  | 6.2        | 6.1  | 6.0  | 5.8  |
>
> Through two sets of experiments, we demonstrate that **CREAM can work seamlessly with PEFT techniques like LoRA [1] and QLoRA [2] without requiring additional modifications**. This compatibility arises because CREAM does not alter any data formats or model structures during fine-tuning; it only adjusts position indices. Detailed experimental results are presented in the following two tables.
>
> | Model              | Single-Doc QA | Multi-Doc QA | Summarization | Few-shot Learning | Code Completion | Synthetic Tasks | Macro |
> | :----------------- | :------------ | :----------- | :------------ | :---------------- | :-------------- | :-------------- | :---- |
> | Llama2-7B-chat-4k* | 24.9          | 22.6         | 24.7          | 60                | 48.1            | 5.9             | 31.0  |
> | LoRA-step-800      | 28.7          | 28.5         | 27.7          | 62.3              |54             |10.3              | 35.3  |
> | QLoRA-step-400     | 20.9          | 19.0         | 26.8          | 54.1              | 47.4            | 4.0             | 28.7  |
> | Ours               | 34.8          | 31.1         | 27.2          | 65.1              | 50.4            | 7               | 35.9  |
>
> We will include the training curves of these two methods in the revised version.
>
> >**W2:** **Lack of Comparative Analysis**: The paper could benefit from a more detailed comparative analysis with other contemporary methods. For example, in Table 2, the paper only compares one model (LongChat-v1.5-7B-32k) with the same 32k context window. I suggest that in such comparision, peer models should at least take the same length of input, otherwise, it cannot prove the effectiveness of the proposed model over other Long LLMs with 32k+ context windows. I suggest the author compare with models like Mistral-7B-Instruct-v0.2, LongLoRA etc.
>
> **A2:** Thank you for the advice for improving our work. Per your suggestion, we have added experimental results of all three versions of Mistral-7B-Instruct (v0.1, v0.2, v0.3) on Longbench. Since the 7B-Instruct-32K model of LongLoRA is not publicly available, we could not add corresponding results and are more than willing to add the comparison once it's open-sourced. The detailed results of Mistral are presented in the table below.
>
> | Model                    | Single-Doc QA | Multi-Doc QA | Summarization | Few-shot Learning | Code Completion | Synthetic Tasks | Macro    |
> | :----------------------- | :------------ | :----------- | :------------ | :---------------- | :-------------- | :-------------- | :------- |
> | Mistral-7B-Instruct-v0.1 | 29.5          | 20.7         | 26.4          | 13.6              | 29.6            | 10.8            | 21.8     |
> | Mistral-7B-Instruct-v0.2 | 28.5          | 21.5         | 26.1          | 50.1              | 33.8            | 13.9            | 29.0     |
> | Mistral-7B-Instruct-v0.3 | 33.2          | 30.6         | 26.8          | 56.4              | 15.3            | 10.4            | 28.8     |
> | LongChat-v1.5-7B-32k*    | 28.7          | 20.6         | 26.7          | 60.0              | 54.1            | 15.8            | 34.3     |
> | **Ours**                 | 34.8          | 31.1         | 27.2          | 65.1              | 50.4            | 7.0             | **35.9** |
>
> As shown, it is evident that **our method outperforms all three versions of Mistral-7B-Instruct**. This further confirms the effectiveness of CREAM.
>
> >**Q1:** What are the computational overheads associated with implementing CREAM, and how does it scale with larger context sizes?
>
> **A1:** Our method **does not incur any additional computational overhead beyond standard fine-tuning**. This is because CREAM does not modify the model structure itself; the only change is to the positional indices. Additionally, as the context window length increases, the computational overhead will increase similarly to standard fine-tuning.
>
> However, even with a larger context window, it is still possible to fine-tune with a 4K context length, significantly reducing computational overhead and memory usage. As as shown in the table in A2 of reviewer srX3's response, fine-tuning LLaMA3-8B with a 4K context length (despite its pre-training context length of 8K) remains remarkably effective.

---

> ### Author Response · Authors · 2024-08-06
> **Rebuttal by Authors**
>
> >**Q2:** Are there any specific hyperparameter settings in CREAM that are particularly sensitive, and how were these chosen?
>
> **A2:** Our method requires only three preset hyperparameters: the mean $\mu$ and variance $\sigma$ of the truncated Gaussian sampling, and the length of the head and tail sections $k$.
>
> For the mean $\mu$, it depends on the expansion factor $\alpha$, specifically $\mu = (1 + \alpha) / 2$.
>
> For the length $k$, we follow the settings of StreamingLLM[3], which uses a value of 4. Since we have expanded by a factor of 8, we set $k = 4 \times 8 = 32$. This also depends on the expansion factor.
>
> For the variance $\sigma$, our setting should ensure higher sampling frequency in the middle and lower (but non-zero) frequency at the ends. The experimental results for different $\sigma$ values in the table in A2 of reviewer fWsk's response can be used as a reference.
>
> >**Limitations:** The discussion in Appendix F regarding the limitations of the CREAM method could be misleading. The author implies that CREAM provides enhanced performance compared to other methods concerning the "Lost in the Middle" issue but acknowledges that the problem is not fully solved due to the nature of decoder-only models. For a genuine limitations section, it is essential to delve deeper into the specific constraints of the CREAM approach and how these limitations might impact its application and effectiveness. A more explicit acknowledgment of the method's shortcomings and a clear explanation of why these issues cannot be overcome with the current model design would significantly improve the section's clarity and usefulness.
>
> **A:** Thank you for noticing our discussed limitations and providing suggestions. We would like to explicitly elaborate the limitations of CREAM as two-fold. First, our method did not involve any adjustments to the model architecture. As discussed in our research question as in L34-35, this work aims to reach an efficient and effective optimality based on a pre-trained model. Consequently, it encounters the decoder-only limit mentioned in [4]. Second, our methods follow prior works and fine-tune pre-trained models on a small dataset considering the efficiency. Nevertheless, our approach might benefit from further enhancement of pre-trained dataset.  We will provide a more detailed discussion of the limitations in our paper.
>
> We hope our answers have resolved your concerns. If you have any other concerns, please feel free to let us know. Thanks again for your review.
>
> Reference:
>
> [1] LoRA: Low-Rank Adaptation of Large Language Models. ICLR, 2022.
>
> [2] Qlora: Efficient finetuning of quantized llms. NeurIPS, 2023.
>
> [3] Efficient Streaming Language Models with Attention Sinks. ICLR, 2024.
>
> [4] Lost in the middle: How language models use long contexts. TACL, 2024.

---

### Official Review · Reviewer_srX3 · 2024-07-13

**Soundness:** 3
**Presentation:** 3
**Contribution:** 2
**Rating:** 6
**Confidence:** 3

**Summary:**

The paper proposes a new method, CREAM, that better enables extrapolation to longer contexts via finetuning at the base context length. The method uses positional embedding interpolation with the embeddings divided into three areas of interest and uses a truncated Gaussian  to sample the position used for the middle segment, to increase training of the middle context. They show that this is at least as effective as prior length extrapolation methods that use finetuning at the base context length, and that CREAM-trained models suffer less from the lost in the middle problem.

**Strengths:**

S1. The method shows clear improvement on the targeted issue (the lost in the middle phenomenon)-- e.g. in Table 1.

S2. The method is well-motivated both intuitively and theoretically, and the explanation in section 2.2 is well-written and easy to follow.

S3. The authors evaluate against the two most reasonable baselines (to my knowledge) and across a good selection of synthetic, long-context, and short-context tasks. (It would be an added benefit to evaluate against finetuning with position interpolation and *long* context at finetuning-time, to see the efficiency-performance tradeoff; however, I do not think this is strictly necessary to the paper's claims.)

**Weaknesses:**

W1. The method is fairly similar to POSE in concept and performance. While I think the idea of emphasizing middle training is reasonable, well-explored, and clearly effective (e.g. in Figure 1), this has also been explored as a post-training or data selection correction in other work. Given the slight performance degradation from POSE on short-context tasks (i.e. in Table 4), I'm concerned that this might not be worth the tradeoff practically to do at training time.

W2. Evaluation settings can differ subtly across papers, and so it would be better for the authors to reproduce baseline results rather than citing them from literature, where possible. This is most critical, I think, in Table 4, where the differences between methods are relatively small and all results are compared to the same baseline method, which is not an infeasibly expensive or unavailable model to run.

**Questions:**

Q1. In Table 5: why does the 256k-extrapolated version of CREAM underperform the 128k-extrapolated version on short contexts?

Q2. Say you had an even more limited amount of compute, to the point where finetuning on the base context length was prohibitively expensive (e.g., maybe you want to finetuned Llama3-8b but can't fit 8192 context at training time). Do you believe CREAM would be useful in this setting? Do you have any results (or speculation!) that suggest when the method may begin to break down with decreasing maximum training length?

Q3. (Minor) What is the context length used in Figure 1? The x axis only reports the position of keys.

Suggestions/comments:
*  Please explain in the text what the occasional highlighting of numbers represents (e.g. in Table 2). It would be helpful for readability to bold the best number in each column across tables.

**Limitations:**

Yes, no concerns here.

---

> ### Author Rebuttal · Authors · 2024-08-06
>
> Thank you very much for your valuable suggestions and acknowledgment of our well-motivated method. Below we address your questions point-by-point and hope we can resolve your major concerns.
>
> >**W1:** The method is fairly similar to POSE in concept and performance. While I think the idea of emphasizing middle training is reasonable, well-explored, and clearly effective (e.g. in Figure 1), this has also been explored as a post-training or data selection correction in other work. Given the slight performance degradation from POSE on short-context tasks (i.e. in Table 4), I'm concerned that this might not be worth the tradeoff practically to do at training time.
>
> **A1:** We would like to highlight the major contributions and differences compared with PoSE, other post-training, and data selection correction as follows.
>
> 1. For the concept, PoSE does not simultaneously leverage the benefits of continuity and relativity, nor does it emphasize the importance of intermediate context. These aspects are our core contributions.
> 2. For the performance, our method **significantly outperforms PoSE across various tasks**. For example, it achieves an average improvement of 15.9% on LongChat Lines compared to PoSE. In the Lost in the Middle task, our different interpolation methods (Linear, NTK, Yarn) also show an average improvement of about 10%.
> 3. In post-training, previous methods usually **require a large amount of data and computational resources, with the training context length being much greater than the context length during pre-training**. For instance, Yi-1.5[1] uses 10 billion tokens and requires upsampling of long sequences. Qwen-2[2] and InternLM2[3] extend their training context length from 4K to 32K in the final stages of pre-training. Therefore, we can conclude that CREAM **not only requires less data and shorter training context lengths but also demonstrates impressive performance**.
> 4. For data selection correction, such as [3] using data engineering techniques to expand context. However, their method requires fine-tuning on at least 500M tokens to achieve satisfactory performance in the Needle-in-a-Haystack test (Figure 3). In contrast, our method only requires $seqlen(4096) \times bsz(32) \times steps(1000) = 125M$ tokens. Therefore, **their training data requirement is not only four times larger than ours but also requires meticulous data selection.** Additionally, **our method can be applied to instruction fine-tuning**, which they did not mention.
>
> It is important to emphasize that on short-context tasks (i.e. Table 4), **our method performs just as well as PoSE**. In our manuscript, we overlooked the minor differences between ours and PoSE in Table 4 (<1%) since our focus and superiority lie in addressing the challenges of long contexts.  The minor performance turbulence results from the sampling of training data (a de facto strategy as in PoSE and others). To further validate this, we tested model performance with five different random seeds used to sample the training data. The mean$\pm$std results are shown in the table below.
>
> | Model           | Zero-Shot  |                 |           |           | Few-Shot  |           |
> | :-------------- | :--------- | :-------------- | :-------- | :-------- | :-------- | :-------- |
> |                 | WinoGrande | TruthfulQA(mc2) | PIQA      | BoolQ     | ARC-C     | HellaSwag |
> | PoSE-Linear  | 68.7±1.30  | 38.6±1.36       | 77.9±0.97 | 76.3±0.74 | 47.4±1.46 | 76.9±0.42 |
> | CREAM-Linear | 68.8±1.30  | 38.5±1.35       | 78.1±0.97 | 76.3±0.75 | 47.5±1.46 | 77.0±0.42 |
>
> Based on the experimental results in the table, we can be more confident that the performance of CREAM and PoSE on short contexts is actually comparable. Thank you for the careful review. We would update and add the discussion in our revision.
>
> >**W2:** Evaluation settings can differ subtly across papers, and so it would be better for the authors to reproduce baseline results rather than citing them from literature, where possible. This is most critical, I think, in Table 4, where the differences between methods are relatively small and all results are compared to the same baseline method, which is not an infeasibly expensive or unavailable model to run.
>
> **A2:** Thank you for your suggestion. We've followed your advice and reproduced the results of LLaMa-2-7b-hf, as presented in Table 4. The results are detailed in the table below:
>
> | Model             | Zero-Shot  |                 |          |          | Few-Shot |           |
> | :---------------- | :--------- | :-------------- | :------- | :------- | :------- | :-------- |
> |                   | WinoGrande | TruthfulQA(mc2) | PIQA     | BoolQ    | ARC-C    | HellaSwag |
> | LLaMa-2-7b-hf*    | 69.2       | 39.5            | 78.8     | 77.4     | 45.9     | 77.2      |
> | **LLaMa-2-7b-hf** | **68.5**   | **38.9**        | **78.8** | **78.0** | **48.5** | **78.1**  |
>
> The bolded results in the table are our reproductions, and they are very close to the original results. Therefore, we believe our final conclusions and claims still stand across all tasks. Thank you for the advice. We will add the above results to improve the soundness of our work.
>
> >**Q1**: In Table 5: why does the 256k-extrapolated version of CREAM underperform the 128k-extrapolated version on short contexts?
>
> **A1:** In Table 5, the 128K results are not ours but are cited from PoSE. Since **a lower perplexity is better**, the results in Table 5 indicate that our method has the potential to extend to 192K or even 256K. Furthermore, compared to PoSE, our performance is significantly superior.

---

> ### Author Response · Authors · 2024-08-06
> **Rebuttal by Authors**
>
> >**Q2:** Say you had an even more limited amount of compute, to the point where finetuning on the base context length was prohibitively expensive (e.g., maybe you want to finetuned Llama3-8b but can't fit 8192 context at training time). Do you believe CREAM would be useful in this setting? Do you have any results (or speculation!) that suggest when the method may begin to break down with decreasing maximum training length?
>
> **A2:** Thank you for providing the experimental setup, which further highlights the superiority of our method. Following your instructions, we fine-tuned LLaMa3-8b using a 4K context window size. The experimental results are presented in the table below.
>
> | AVG Length          | 2000 | 2700 | 3300 | 4000 | 5200 | 6500 | 7800 | 8800 | 9700 | 11000 | 12000 | 14000 | 17000 | 19000 | 24000 | 28000   | 32000 |
> | :------------------ | :--- | :--- | :--- | :--- | :--- | :--- | :--- | :--- | :--- | :---- | :---- | :---- | :---- | :---- | :---- | :------ | :---- |
> | LLaMa3-CREAM-Linear | 0.98 | 0.96 | 0.98 | 1.00  | 0.92 | 0.96 | 0.96 | 0.94 | 0.86 | 0.92  | 0.92  | 0.92  | 0.86  | 0.84  | 0.70   | 0.60 | 0.48  |
>
> The results presented in the table demonstrate that our method **is well-suited to this setting and performs surprisingly well.**
>
> >**Q3:** (Minor) What is the context length used in Figure 1? The x axis only reports the position of keys.
>
> **A3:** This is consistent with the setting in [5] and corresponds to an approximate length of 5K. Results with longer contexts (10K) are in Table 1.
>
> >**Suggestions/comments:**
> >Please explain in the text what the occasional highlighting of numbers represents (e.g. in Table 2). It would be helpful for readability to bold the best number in each column across tables.
>
> **A:** Thank you for the insightful suggestions. The highlighted numbers in Table 1 and Table 2 emphasize the advantages of our results compared to other methods. In Table 2, we have bolded the average results. Following your suggestion, we will also bold the best results in each column.
>
> We hope our answers have resolved your concerns. If you have any other concerns, please feel free to let us know. Thanks again for your review.
>
> Reference:
>
> [1] Yi: Open foundation models by 01. arXiv preprint arXiv:2403.04652, 2024.
>
> [2] Qwen2 technical report. arXiv preprint arXiv:2407.10671, 2024.
>
> [3] Internlm2 technical report. arXiv preprint arXiv:2403.17297, 2024.
>
> [4] Data Engineering for Scaling Language Models to 128K Context. ICML, 2024.
>
> [5] Lost in the middle: How language models use long contexts. TACL, 2024.

---

> > ### Comment · Reviewer_srX3 · 2024-08-12
> >
> > Thanks for the detailed response!
> >
> > > W1
> >
> > I appreciate the multiple random seeds for the short-context performance eval (and would love to see this extended to report error bars on the long-context evals as well). My major concern was short-context regression relative to POSE, and looking at the results over multiple random seeds shows that there is not a significant difference in short-context performance.
> >
> > > Q1
> >
> > Totally my bad, somehow missed that this table was perplexity!
> >
> > > we fine-tuned LLaMa3-8b using a 4K context window
> >
> > I'm also glad to see how well the method performs in this even more length-constrained scenario.
> >
> > Given the rebuttal, I have raised my score 5->6. Thanks for the nice paper!

---

> > > ### Author Response · Authors · 2024-08-13
> > >
> > > We would like to express our sincere gratitude for your thorough review of our paper. Your expertise has greatly contributed to enhancing the quality of our work, and we are committed to incorporating your suggestions during the revision process. Thank you once again for acknowledging our efforts.

---

### Official Review · Reviewer_fWsk · 2024-07-13

**Soundness:** 2
**Presentation:** 4
**Contribution:** 4
**Rating:** 6
**Confidence:** 3

**Summary:**

The paper proposes a method for extending the context window of pretrain large language models. The approach, CREAM, relies on modifying position indices to interpolate the positional encodings. Despite the often computationally expensive nature of such work, CREAM can extend to very long context windows while only needing to train at the original pretrained context window. Additionally, in their approach to the context length problem, the authors propose a solution that focuses explicitly on learning at the middle of the context — a span that often under-performs in long context models. The experimental results cover a wide variety of problems, both in terms of tasks, as well as context length challenges. Overall the results look very promising, and show a strong method for addressing a challenging task of extending the context length of LLMs.

**Strengths:**

- The authors tackle two problems that go hand in hand, namely extending the context length during fine-tuning - but doing so in a way ensures consistent performance.
- The approach divides the desired context length into three segments, which then results in relative positional distances that vary and consequently learning all relative positions within the target length $L$. The technique is simple, but clever and an effective way to efficiently expose the model to a broader range of relative positional distances during training.
- The results are very strong, particularly with the performances listed on Long Bench, which encompasses a broad range of tasks.

**Weaknesses:**

- No error bars shown on results. In most cases the results are quite strong, but the error bars would be helpful — particularly in some of the closer comparisons with PoSE.
- The solution of using a truncated Gaussian approach lacks motivation. The "lost in the middle" problem is clear, however the solution of using a truncated Gaussian to force more focus on the middle seems brittle. Could the parameters of the truncated Gaussian be learned from data? The approach works well based on the results shown in Figure 5a, however.

**Questions:**

The authors mention that CREAM only needed to be trained for 100 in some cases. Can the authors provide more insight on why the results are very good despite so little training? This seems like a significant achievement, but was not discussed in detail.

**Limitations:**

The amount of focus on the middle context seems fixed in this approach. While the results for Gaussian truncation appear promising, it's not clear if this direct approach to solving the "lost in the middle" problem is universal. In short, how do we know how much to reweight the importance of the middle context, and does it vary by dataset or task?

---

> ### Author Rebuttal · Authors · 2024-08-06
>
> We sincerely thank you for your constructive comments ``simple, but clever, effective and effcient`` and acknowledgement of our approach with ``very strong performances that encompass a broad range of tasks``. Below, we provide detailed replies to your comments and hope we can resolve your major concerns.
>
> >**W1**：No error bars shown on results. In most cases the results are quite strong, but the error bars would be helpful — particularly in some of the closer comparisons with PoSE.
>
> **A1:** Thank you for the helpful suggestion. In our manuscript, we fixed all random seeds of all experiments to guarantee the reproducibility of our reported results. Per your advice, we have further conducted repeated experiments with five randomly generated seeds on the linear extension experiment. The detailed results with mean$\pm$std are reported in the table below.
>
> | Model           | Zero-Shot  |                 |           |           | Few-Shot  |           |
> | :-------------- | :--------- | :-------------- | :-------- | :-------- | :-------- | :-------- |
> |                 | WinoGrande | TruthfulQA(mc2) | PIQA      | BoolQ     | ARC-C     | HellaSwag |
> | PoSE-Linear  | 68.7±1.30  | 38.6±1.36       | 77.9±0.97 | 76.3±0.74 | 47.4±1.46 | 76.9±0.42 |
> | CREAM-Linear | 68.8±1.30  | 38.5±1.35       | 78.1±0.97 | 76.3±0.75 | 47.5±1.46 | 77.0±0.42 |
>
> By comparing the mean and variance in the table, it can be observed that CREAM and PoSE have comparable capabilities in handling short texts. We will add the results in our revision.
>
> >**W2:** The solution of using a truncated Gaussian approach lacks motivation. The "lost in the middle" problem is clear, however the solution of using a truncated Gaussian to force more focus on the middle seems brittle. Could the parameters of the truncated Gaussian be learned from data? The approach works well based on the results shown in Figure 5a, however.
>
> **A2:** Thank you for pointing this out. Our motivations for truncated Gaussian are three-fold:
>
> 1. **Intuitive Explanation.** The issue of "lost in the middle" highlights that the performance of LLMs is often strong at the beginning and end, but weak in the middle[1].  As discussed in **L111-114**, a straightforward idea to address this is to **guide LLMs to focus more on the middle part relative to the beginning and end** of the context. This approach results in a "reverse U" shape curve, similar to a truncated Gaussian distribution curve.
> 2. **Theoretical support.** In Appendix B and L114-115, we provide theoretical justification that sampling the middle part of a long context with a high importance rate yields a maximization of learned relative position intervals as identified in Equations (2).
> 3. **Empirical observation.** As you mentioned, it has proven to be quite effective for all demonstrated long context tasks, encompassing retrieval, lost-in-the-middle, LongBench, etc.
>
> **Could the parameters of the truncated Gaussian be learned from data?**
>
> Great question. In our truncated Gaussian sampling, as shown in Equation (3), the only hyperparameters are the mean $\mu$ and the variance $\sigma$.
>
> 1. For $\mu$, it depends on the expansion factor. For example, when expanding from 4K to 32K, the expansion factor is 8, so $\mu = (1+8)/2$.
>
> 2. For $\sigma$, learning from the data is flexible, but **since the sampling process is discrete, the gradient cannot be back-propagated, making the learning cost high.** Additionally, we conducted experiments with five different $\sigma$ values, and the results in the table show that the current choice ($\sigma=3$) indeed performs the best.
>
> | Length    | 2500 | 3600 | 4200 | 4800 | 6000 | 7100 | 9400 | 11800 | 14000 | 16000 | 17500 | 20000 | 22000 | 26000 | 28000 | 30000 | 32000 | AVG   |
> | :-------- | :--- | :--- | :--- | :--- | :--- | :--- | :--- | :---- | :---- | :---- | :---- | :---- | :---- | :---- | :---- | :---- | :---- | :---- |
> | sigma=2   | 0.68 | 0.56 | 0.7  | 0.66 | 0.7  | 0.68 | 0.68 | 0.56  | 0.52  | 0.4   | 0.42  | 0.38  | 0.48  | 0.38  | 0.4   | 0.28  | 0.12  | 0.506 |
> | sigma=2.5 | 0.9  | 0.72 | 0.78 | 0.86 | 0.78 | 0.76 | 0.7  | 0.56  | 0.64  | 0.38  | 0.52  | 0.4   | 0.48  | 0.36  | 0.46  | 0.34  | 0.3   | 0.585 |
> | sigma=3   | 0.96 | 0.82 | 0.92 | 0.94 | 0.92 | 0.86 | 0.84 | 0.78  | 0.76  | 0.56  | 0.62  | 0.52  | 0.62  | 0.46  | 0.52  | 0.38  | 0.4   | 0.699 |
> | sigma=3.5 | 0.9  | 0.72 | 0.84 | 0.8  | 0.84 | 0.78 | 0.74 | 0.66  | 0.58  | 0.4   | 0.54  | 0.4   | 0.5   | 0.38  | 0.38  | 0.36  | 0.26  | 0.593 |
> | sigma=4   | 0.9  | 0.8  | 0.86 | 0.84 | 0.78 | 0.72 | 0.72 | 0.5   | 0.5   | 0.42  | 0.44  | 0.26  | 0.36  | 0.22  | 0.3   | 0.3   | 0.26  | 0.540 |

---

> ### Author Response · Authors · 2024-08-06
> **Rebuttal by Authors**
>
> >**Q:** The authors mention that CREAM only needed to be trained for 100 in some cases. Can the authors provide more insight on why the results are very good despite so little training? This seems like a significant achievement, but was not discussed in detail.
>
> **A**: Thank you for the careful review. For the Base model (Llama2-7b), we conducted 1,000 steps of continual pre-training on the to ensure a fair comparison with prior works as in [5,6]. For the Chat model (Llama2-7b-chat), we performed only 100 steps of Instruction Fine-Tuning (IFT).  Indeed, we attempted to extend the IFT to 400 steps, but the performance deteriorated; the detailed results are shown in the table below. This is because we used the instruction dataset ShareGPT as a substitution for the original IFT used for Chat model (the original dataset is not publicly accessible). This observation is in line with previous studies that have pointed out that **instruction fine-tuning on different datasets may lead to catastrophic forgetting and impact performance**[2,3,4]. To balance the effectiveness of position index learning and the degradation of the LLM's IFT performance, we limited the IFT to 100 steps. Like you have observed, 100 steps of finetuning yield effective long-context performance.
>
> | Model              | Single-Doc QA | Multi-Doc QA | Summarization | Few-shot Learning | Code Completion | Synthetic Tasks | Macro |
> | :----------------- | :------------ | :----------- | :------------ | :---------------- | :-------------- | :-------------- | :---- |
> | Llama2-7B-chat-4k* | 24.9          | 22.6         | 24.7          | 60                | 48.1            | 5.9             | 31.0  |
> | 100 steps          | 34.8          | 31.1         | 27.2          | 65.1              | 50.4            | 7.0             | 35.9  |
> | 400 steps          | 30.9          | 23.5         | 27.1          | 62.4              | 35.6            | 3.4             | 30.5  |
>
> >**Limitations:** The amount of focus on the middle context seems fixed in this approach. While the results for Gaussian truncation appear promising, it's not clear if this direct approach to solving the "lost in the middle" problem is universal. In short, how do we know how much to reweight the importance of the middle context, and does it vary by dataset or task?
>
> **A:** Sorry about confusion. We would like to clarify that **the middle context our method focuses on is NOT fixed**. According to Algorithm 1, **each sample requires two rounds of sampling to obtain the final position index, so the intermediate context position index varies between samples.** Specifically, we first sample a factor $\alpha$ from a truncated Gaussian distribution to determine the interval where the current $P_e$ is located. Then, we uniformly sample the final $P_e$ from this interval, which is the ending position index of the intermediate section. The sample flow was discussed in L122-125. We would further add more clarification to explain our strategy.
>
> As mentioned above, our method **does NOT require meticulously designed weight distributions**; instead, it automatically assigns weights using truncated Gaussian sampling. Moreover, in all the experiments presented in our paper, all parameters remained consistent, yet the method still performed effectively across various tasks. This demonstrates the generality of our approach, which **does not change depending on the dataset or task**.
>
> We hope the above response can resolve your questions and concerns. Please let us know if there is any further question! Thanks again for your review.
>
> Reference:
>
> [1] Lost in the middle: How language models use long contexts. TACL, 2024.
>
> [2] CoachLM: Automatic Instruction Revisions Improve the Data Quality in LLM Instruction Tuning. ICDE, 2024.
>
> [3] Lima: Less is more for alignment. NeurIPS, 2024.
>
> [4] From Quantity to Quality: Boosting LLM Performance with Self-Guided Data Selection for Instruction Tuning. NAACL 2024.
>
> [5] Extending Context Window of Large Language Models via Positional Interpolation
>
> [6] PoSE: Efficient Context Window Extension of LLMs via Positional Skip-wise Training, ICLR 2024.

---

> > ### Comment · Reviewer_fWsk · 2024-08-14
> >
> > Thank you for the response. You have answered each of my questions and provided clarity on some points I misunderstood. I appreciate the additional analyses you’ve run — they provide insight and more confidence in your findings.

---

> > > ### Author Response · Authors · 2024-08-14
> > >
> > > Thank you for your thorough review of our rebuttal and your encouraging feedback. Your expertise has significantly contributed to improving the quality of our work, and we are dedicated to incorporating your suggestions in our revision process. If you are satisfied with our rebuttal, we would be extremely grateful if you could consider increasing the final score.

---

### Author Rebuttal · Authors · 2024-08-06

We appreciate all the reviewers for their hard work, and we will re-emphasize a few strengths of our work:

1. We propose a ``simple, but clever and an effective way to efficiently expose the model to a broader range of relative positional distances during training`` (Reviewer fWsk). This ``novel positional encoding strategy that efficiently extends the context window of LLMs, which is a significant contribution to the field`` (Reviewer XDBh).
2. Our method ``provides strong empirical evidence through comprehensive experiments, demonstrating CREAM's superiority over existing methods like PoSE and RandPos, especially in handling middle context information``  (All Reviewers) . Additionally, our method ``is computationally efficient compared to fine-tuning at the target length`` (Reviewer XDBh).
3. Notably, our ``method is well-motivated both intuitively and theoretically, and the explanation in section 2.2 is well-written and easy to follow`` (Reviewer srX3,XDBh).

In the subsequent revisions, we address all the reviewers' comments by making the following modifications to our paper:

1. We add the surprisingly good experimental results of fine-tuning LLaMa3-8B using CREAM on 4K data length. Additionally, we include the experimental results of applying CREAM on Baichuan2-7B to demonstrate the generality of our approach across different scenarios and LLMs. (Reviewers srX3, XDBh)
2. We include the experimental results of combining CREAM with different PEFT techniques, such as LoRA and QLoRA, to further prove the versatility of our method. (Reviewer XDBh)
3. We add the ablation results of different $\sigma$ values. (Reviewers fWsk, XDBh)
4. We include error bars in the results of Table 4 and add the reproduced results of LLaMa2-7B. (Reviewers fWsk, srX3)
5. We add the experimental results of Mistral-7B-Instruct (v0.1, v0.2, v0.3) on Longbench in Table 2 to further validate the effectiveness of our approach. (Reviewer XDBh)
6. We further discuss the reasons for the limitations of our method in the limitations section. (Reviewer XDBh)

We hope our responses will address the reviewers' concerns. If there are any questions, we look forward to further discussions.

---

### Decision · Program_Chairs · 2024-09-25

**Decision:**

Accept (poster)

**Comment:**

The paper introduces CREAM, a method for extending the context window of large language models that emphasizes middle context enhancement, showing promising results with efficient training and broad applicability.  CREAM manipulates position indices and employs a truncated Gaussian distribution to focus on the middle part of the context, effectively addressing the "Lost-in-the-Middle" problem.  The proposed method demonstrates superior performance over existing methods like PoSE and RandPos across various tasks. The method shows robustness and has the potential to significantly enhance LLMs' ability to process long contexts.  The authors well addressed the reviewers' concerns in the rebuttal.  Reviewers are overall positive about the paper except for one who suggests to reject but seems not very confident.